# Enhance Reasoning for Large Language Models with Reinforcement Learning in the Game Werewolf

## Abstract

Despite their success across a broad spectrum of general tasks, Large Language Models (LLMs) often underperform in domain-specific tasks not well-represented in their pre-training corpora. We introduce an innovative framework integrating general-purpose LLMs with an external *Thinker* module to enhance the reasoning capabilities of LLM-based agents. Unlike augmenting LLMs with prompt engineering, our Thinker module directly accesses knowledge from domain databases and employs supervised or reinforcement learning (RL). We establish a reasoning hierarchy where LLMs handle intuitive *System-1* tasks that are domain-agnostic, while the Thinker focuses on *System-2* tasks that require complex logical analysis and domain-specific knowledge. Our framework is demonstrated through a 9-player Werewolf game that necessitates dual-system reasoning. We design a communication protocol between LLMs and the Thinker, then optimize the Thinker through online RL and refine it by imitation learning. Drawing from 18 800 human games, this work also contributes to the largest dataset for social deduction games to date. Experiments show that GPT-3.5 and GPT-4, augmented with the Thinker, significantly improve in deductive reasoning, textual speech generation, and online gameplay evaluated by human players. Further, integrating a fine-tuned 6B Werewolf-specific LLM with the Thinker achieves performance on par with GPT-4.

## 1 Introduction

The field of artificial intelligence has witnessed groundbreaking advancements in recent years, with the development of Large Language Models (LLMs) (Ouyang et al., 2022; OpenAI, 2023; Anil et al., 2023). Apart from their impressive proficiency in natural language processing (NLP) tasks (Thoppilan et al., 2022; Zhang et al., 2023b), LLMs also exhibit vast potential as general problem solvers in areas such as planning and decision-making (Huang et al., 2022), knowledge transfer and generalization (Anil et al., 2022) and multi-modal perception (Yin et al., 2023) due to the rich world knowledge embedded in their training corpora. Consequently, the integration of LLMs as central controllers with task agents for end-to-end solutions has emerged as a promising research direction, yielding breakthroughs in domains including tools and assistants (Schick et al., 2023; Ge et al., 2023), engineering (Ahn et al., 2022), and gaming (Wang et al., 2023).

LLM-based agents utilize LLMs for their general-purpose reasoning abilities (Huang & Chang, 2022), which are primarily enabled by prompt engineering methods such as information profiling (Zhang et al., 2023a; Qian et al., 2023), step-by-step task decomposition (Wei et al., 2022b; Zhou et al., 2022), recursive prompting by feedback from the environment (Yao et al., 2022), human interaction (Wu et al., 2022) and self-refinement (Madaan et al., 2023; Shinn et al., 2023). These methods obviate the need for domain-specific fine-tuning of LLMs. To augment their task-specific competencies, researchers adopt external modules like memory systems for storing and retrieving historical information (Lin et al., 2023; Zhong et al., 2023; Hu et al., 2023), external tools (Schick et al., 2023), APIs (Qin et al., 2023), knowledge bases (Lewis et al., 2020) and expert models (Yang et al., 2023b; Ge et al., 2023).

Despite these advancements, challenges persist in domain-specific applications, where LLM-based agents often serve primarily as demonstrations rather than as practical solutions (Yang et al., 2023a;

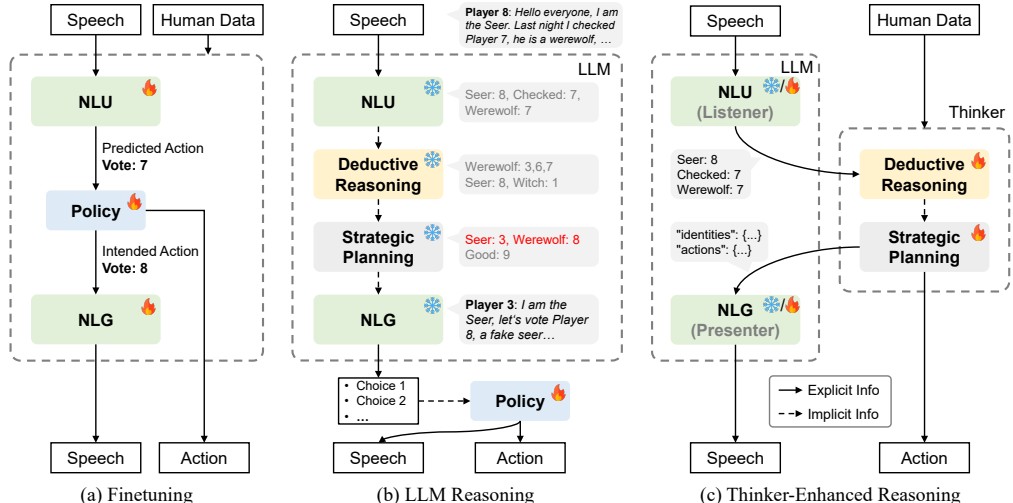

Figure 1: Comparing related approaches. (a) Alignment by fine-tuning of LLMs (Bakhtin et al., 2022); (b) Reasoning mainly by LLMs (Xu et al., 2023b); (c) A dual-system reasoning hierarchy of LLMs and the Thinker. Snow and fire represent without/with finetuning of the model.

Tan et al., 2024). First, while general-purpose LLMs have emerged some reasoning capabilities, they require sufficient model scales and computational overheads (Kaplan et al., 2020), along with various aforementioned techniques (Wei et al., 2022a). However, LLMs struggle to achieve satisfactory performance in higher-level reasoning (Stechly et al., 2023; Dziri et al., 2023) and planning (Valmeekam et al., 2023; Bubeck et al., 2023) tasks. Second, most LLM-based agents avoid fine-tuning LLMs on task-specific data to preserve the model's generality and prevent over-fitting. This strategy complicates the utilization of existing domain datasets and expertise, as well as the alignment of task scenarios with input-output formats, data distributions, and human preferences.

Addressing the limitations of LLMs in high-level and domain-specific reasoning, we draw inspiration from the dual-process theory (Wason & Evans, 1974; Daniel, 2017; Hagendorff et al., 2022; Weston & Sukhbaatar, 2023; Lin et al., 2024) and distinctly separate reasoning into two systems. We propose an external *Thinker* module to enhance the reasoning capabilities of LLMs, as shown in Figure 1(c). In our framework, LLMs are responsible for *System-1* reasoning related to intuitive thinking, such as domain-agnostic NLP interactions, common-sense and symbolic reasoning, while the Thinker handles *System-2* reasoning that is deliberate, analytical, and requires deep understanding of domain-specific knowledge. We design a communication protocol between LLMs and the Thinker through explicit information. Unlike augmenting LLMs with prompt engineering, the Thinker directly accesses knowledge from extensive databases and applies various optimization techniques, thus enhancing the performance and human alignment without compromising LLM's generality.

The 9-player Werewolf game serves as the proving ground for our framework, given that current AI systems lag significantly behind even moderately skilled human players in this popular social deduction game. We heuristically dissect the reasoning process into four stages, as illustrated in Figure 1(b). *System-1* reasoning includes natural language understanding (NLU) and generation (NLG) of players' speech. Meanwhile, the hidden identities require complex **deductive reasoning** and **strategic planning** such as deception and disguise, which fall under *System-2* reasoning. This duality creates a significant gap between the players' actual speeches and their true intentions, making Werewolf an ideal testbed for assessing advanced reasoning capabilities of LLM-based agents.

We identify primary patterns from real human speeches and design language-based features (from LLMs to Thinker) and speech instructions (from Thinker to LLMs) accordingly. The Thinker is optimized by imitation learning, reinforcement learning (RL) from fictitious self-play (Heinrich et al., 2015), and population-based training (Jaderberg et al., 2017), to produce reasonable and human-aligned game actions and instructions. We compare our approach to the Least-to-Most (LtM) prompting (Zhou et al., 2022) and 11-shot in-context learning from three dimensions: accuracy of deductive reasoning, human preference of generated speeches, and online evaluation of a complete game. Experiments show that the integration of the Thinker module substantially enhances the

reasoning and generation capability of GPT-3.5 and GPT-4. Further, we fine-tune a 6B (Du et al., 2021) werewolf-specific LLM to better align human speech styles. When augmented with the Thinker, it achieves performance comparable to that of GPT-4. Our primary contributions include:

- We propose an innovative Thinker module designed to enhance the reasoning capabilities of LLMs, demonstrated through a Werewolf AI that outperforms GPT-4 with prompt engineering.
- We collect and release a comprehensive dataset[1] of 18800 real human Werewolf game sessions, which represents the largest dataset for social deduction games to date.

## 2 RELATED WORK

**Enhance Reasoning in LLMs.** Several approaches bypass the intricacies of prompt engineering. For instance, LLM+P (Liu et al., 2023) employs an external planner to tackle long-horizon robot planning challenges. A different approach (Zhang et al., 2023a) heuristically designs a low-level planner to manage primitive control actions. The Retrieval-Augmented Generation (RAG) (Lewis et al., 2020) merges pre-trained parametric memory generation models with non-parametric memory, aiming to enhance performance in knowledge-intensive tasks. Regarding the fine-tuning of LLMs, Galactica (Taylor et al., 2022) is trained on a scientific dataset that emphasizes detailed reasoning processes. WebGPT (Nakano et al., 2021) utilizes human feedback to fine-tune GPT-3, enabling it to answer long-form questions within a textual web-browsing context. Toolformer (Schick et al., 2023) fine-tunes LLMs for using external tools in a self-supervised manner with human demonstrations. Swiftsage (Lin et al., 2024) employs GPT-4 with a small LM finetuned on the oracle agent's action trajectories. DECKARD (Nottingham et al., 2023) trains an RL agent to execute subgoals planned by LLMs in the game Minecraft. OpenAGI (Ge et al., 2023) implements RL from feedback in open-ended tasks to refine the LLM's planning strategy. Cicero (Bakhtin et al., 2022) fine-tunes LLMs to generate dialogue controlled by a strategic reasoning module in the game Diplomacy, as shown in Figure 1(a). Our approach diverges from Cicero in several key aspects: the predicted/intended actions in Cicero (1) require both NLU and NLG involves a high-level and task-related reasoning beyond domain-agnostic NLP; (2) necessitate fine-tuning of LLMs; (3) are insufficient to convey complex language dynamics in the Werewolf game (see Appendix B.1 and B.2).

**AI for Social Deduction Games.** DeepRole (Serrino et al., 2019) combines counterfactual regret minimization (CFR) with deep value networks in the non-speech 5-player Avalon game. Hidden Agenda (Kopparapu et al., 2022) presents a two-team, non-speech social deduction game in a 2D environment. A system comprising three LLM-powered interfaces is created (Zhu et al., 2023) to aid gameplay in Dungeon Master. Regarding AI for Werewolf games, bootstrap aggregating and weighted ensemble learning have been applied to refine voting strategies (Khan & Aranha, 2022). Brandizzi et al. (2021) proposes an RL framework to analyze the influence of diverse communication behaviors among agents. One Night Ultimate Werewolf (Eger & Martens, 2019) explores human responses to various deliberation strategies. In the 5-player werewolf game, Wang & Kaneko (2018) builds a deep-Q network to decide whom to trust or kill. Deep Wolf (Shibata et al., 2023) fine-tunes a RoBERTa-like pretrained model with 48 game logs to construct a value network given the current game state, human speeches, and candidate actions. The 7-player version is explored with RL and LLMs in Xu et al. (2023b;a). Our approach differs from previous studies in two fundamental ways: First, the Thinker separates LLMs from domain-specific reasoning. In contrast, LLMs in Xu et al. (2023b) tackle most reasoning tasks and generate candidate outcomes from which an RL model selects to mitigate biases, as illustrated in Figure1(b). Second, by collecting and leveraging authentic human sessions and speech data, we aim for closer alignment with real-world scenarios and human patterns beyond the few-shot capabilities of general-purpose LLMs.

## 3 METHODS

We introduce an innovative framework that synergizes LLMs with an external Thinker module for reasoning and decision-making. To facilitate communication between the Thinker and LLMs, we propose a protocol utilizing language-based features and speech instructions. The framework is thus decomposed into three processing modules:

---

[1] https://anonymous.4open.science/r/werewolf-1B74

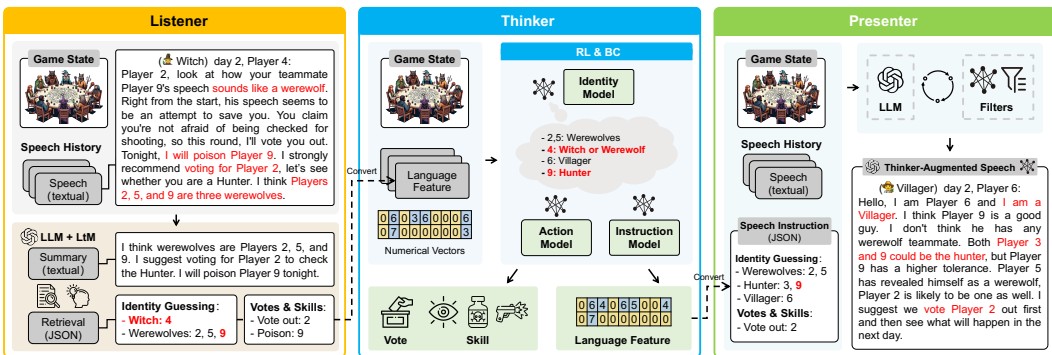

Figure 2: The overall framework and its processing modules in the Werewolf implementation. The retrieval results from the Listener and the speech instructions for the Presenter are formatted in JSON-style for LLMs, while the language features generated by the Thinker are represented as numerical vectors.

- The **Listener** focuses specifically on domain-agnostic NLU tasks. It summarizes lengthy contexts, retrieves key information from natural language inputs, and transforms it into structured language features that the Thinker can interpret.

- The **Thinker** serves as the cognitive core of the framework. Utilizing language features provided by the Listener, it specializes in *System-2* reasoning tasks that require deep logical analysis and domain-specific knowledge. The Thinker produces policies such as planning and actions, and generates strategic instructions for the Presenter.

- The **Presenter** functions as the framework's articulator. Augmented by the strategic instructions from the Thinker, it generates coherent and contextualized language output that is logical, rational, consistent, free from hallucinations, and aligns with the current environment state.

It is important to note that the Listener and Presenter are separated functionally but can be instantiated by a single LLM. Therefore, the framework essentially comprises only an LLM and a Thinker module interacting with each other. To demonstrate the effectiveness of the framework, we apply it to the complex social deduction game Werewolf. The remainder of this section will detail the implementation, which necessitates deductive reasoning, speech understanding and generation, as illustrated in Figure 2.

## 3.1 DATA PREPARATION

We collected data from the 9-player standard mode Werewolf game hosted on the Fanlang platform[2]. The specific rules of the game are detailed in Appendix C. We recorded real-time video in spectator mode for approximately 18 800 game sessions, which equates to around 7000 hours of gameplay and 6000 hours of audio. Furthermore, we enriched our dataset with a Werewolf domain-specific corpus comprising nearly 1.4 million characters, derived from web-crawled game strategies and OCR-processed Werewolf literature. Each recorded session includes both the game state data and the audio of players' speeches. We captured exhaustive game details, such as historical skill usage and voting results, by utilizing the Android automated testing framework[3]. The Paraformer (Gao et al., 2022) model was deployed for Automatic Speech Recognition (ASR) of human speech audio. To improve recognition accuracy, especially for frequently used terms, we developed a list of hot words from the Werewolf corpus and applied context biasing methods (Zhao et al., 2019). Furthermore, we annotated approximately 127 hours of Werewolf speech data and performed supervised fine-tuning on the Paraformer model. The character error rate of ASR for Werewolf speeches was reduced from 4.5% to 3.7%. We refer to the dataset hereafter as *FanLang-9*, with a thorough analysis provided in Appendix D.

---

[2]https://www.wolfkills.com/
[3]https://github.com/appium/appium

## 3.2 LISTENER FOR DOMAIN-AGNOSTIC RETRIEVAL

The complexity of social deduction games stems from players concealing their identities. In the game of Werewolf, Werewolves disguise themselves as members of the "Good" faction through deceptive statements. Conversely, the "Good" faction strives to discern Werewolves by deducing from historical speeches and actions while providing rational and credible statements. This interplay significantly widens the gap between what players say and their true intentions (see Figure 5). The Listener aims to capture relevant insights from actual speeches without speculating on their hidden motives or truthfulness. To address these challenges, we introduce a dual-phase processing:

**Summary**: Human players' speeches on the Fanlang platform are characterized by an information overload. This includes a tangled mix of context, lengthy and redundant content, and colloquial ramblings. Additionally, the speeches feature complex logic that encompasses quotations, rhetorical questions, hypotheses, and empathetic thinking. Together, these elements result in a rich and intricate web of discourse, the accumulation of historical speeches often exceeds 10k tokens (see Figure 9), making it difficult for LLMs to directly retrieve key information from raw contexts. Inspired by the Least-to-Most (LtM) prompting (Zhou et al., 2022), we prompt LLMs to generate a textual summary not exceeding 200 words for each single speech, retaining only critical information that the speaker intends to express.

**Retrieval**: Then we allow the same LLM to retrieve key information from the summary and generate a JSON-style reasoning result given 10 examples, which represents description of players' attributes in the speech. Finally, the result is tokenized and categorized into language features according to specific patterns in Table 8. For an $N$-player Werewolf game, we define $M$ different attributes, which encompass various aspects of a player mentioned in the speech, e.g., identity guessing, historical or future skills and voting decisions. From the historical collection of all speeches $\mathcal{H}$, a player's single speech $\mathbf{S}$ may include descriptions of all the players in the game, the language feature can be represented as a matrix $\mathbf{F} \in \mathbb{Z}^{N \times M}$:

$$\mathbf{F} = [\mathbf{f}_1, \mathbf{f}_2, ..., \mathbf{f}_N]^T, \tag{1}$$

where $\mathbf{f_n} = [f_{n1}, f_{n2}, ..., f_{nM}]^T, n = 1, 2, ..., N$ and $f_{nm} \in \mathcal{V}_m, \forall n = 1, 2, ..., N$ and $m = 1, 2, ..., M$. Here $\mathcal{V}_m$ signifies the set of the potential values that the $m$-th attribute can assume.

An example of summary and language feature ($N = 9, M = 2$) is illustrated in Figure 2. It is worth noting that the dimensions of language features are significantly richer than the predicted/intended actions outlined in Cicero (Bakhtin et al., 2022). Ablation studies (Appendix B.1 and B.2) demonstrate that using actions as compressed representation of speeches leads to substantial information loss and performance degradation in the Werewolf game. Aside from directly prompting LLMs to generate language features, we also extract 260k speech instances from the *FanLang-9* dataset, label the speech-feature pairs with GPT-3.5, and fine-tune a ChatGLM-6B (Du et al., 2021) model, named as WereLLM, to perform the same task for practical efficiency. The detailed prompts for summary and retrieval, as well as the details for fine-tuning of the Listener, are provided in Appendix F.7 and F.6, respectively.

## 3.3 THINKER FOR DOMAIN-SPECIFIC REASONING AND PLANNING

The Listener extracts key information from speech contents to generate language features. Then, the Thinker, utilizing the game state and all historical language features, deduces the underlying intentions and strategic implications of players' public speeches. For example, as shown in Figure 2, although Player 4 claims to be the Witch and accuses Player 9 of being a Werewolf, from Player 6's perspective, Player 4 might be a Werewolf disguising as the Witch, and Player 9 could more likely be the Hunter. Subsequently, it plans game-related actions and speech instructions.

The speech instruction $\mathbf{I} \in \mathbb{Z}^{N \times M}$ follows the same structure as the language feature in Equation 1, except that it is presented in JSON-style to align LLM input. This format is consistent with the retrieval results in the Listener. The generation of a speech instruction can be viewed as a multi-label classification problem and decomposed into multiple single-class classifications for each attribute $f_{nm}$. Therefore, we convert it into $N \times M$ discrete actions and apply the identical training algorithm used for game actions. The optimization of the Thinker comprises two phases: imitation learning and RL. For the imitation learning phase, we utilize human data and employ the Behavioral Cloning

(BC) (Torabi et al., 2018) loss as:

$$\mathcal{L}_{\mathrm{BC}}(\theta) = -\mathbb{E}_{s,a\sim\mathcal{D}}[\log \pi_\theta(a|s)], \qquad (2)$$

where $\mathcal{D}$ denotes the dataset of human action $a$ (or decomposed speech attribute $f_{nm}$), state $s$, and $\pi_\theta$ is the policy parameterized by $\theta$. Since there are some gaps between the *FanLang-9* dataset and our simulation environment (see Appendix C.3), we further apply RL phase utilizing Proximal Policy Optimization (PPO) (Schulman et al., 2017) and a distributional training framework (Ye et al., 2020):

$$\mathcal{L}_{\mathrm{RL}}(\theta) = -\mathbb{E}_{s,a\sim\pi_{\theta'}}\left[\frac{\pi_\theta(a|s)}{\pi_{\theta'}(a|s)}A^{\pi_\theta}(s,a)\right], \qquad (3)$$

where $\theta'$ is the parameters of an old policy, and $A^{\pi_\theta}(s,a)$ is the advantage with respect to policy $\pi_\theta$, which is calculated by the Generalized Advantage Estimator (GAE) (Schulman et al., 2015). Additionally, we integrate an identity model designed to predict the identities of all players, which uncovers the Thinker's real deductions and may diverge from speech instructions it generates. We denote the loss function as $\mathcal{L}_{\mathrm{id}}(\phi)$ with parameter $\phi$, which is labeled by the game environment in a self-supervised manner. The overall training objective of the Thinker is formulated as:

$$\mathcal{L} = \mathcal{L}_{\mathrm{RL}}(\theta) + \alpha\mathcal{L}_{\mathrm{BC}}(\theta) + \beta\mathcal{L}_{\mathrm{id}}(\phi), \qquad (4)$$

where $\alpha$ and $\beta$ are weighting coefficients.

During the Thinker's training, we assume that the Presenter generates speech accurately based on the speech instructions, and the Listener processes this speech and generate a language feature that precisely matches the original speech instruction. This allows speech instructions to be directly regarded as language features, thus enabling the **Thinker to be optimized independently of the Listener and Presenter**. Given the game's asymmetric and adversarial nature, maintaining a balanced win rate between the two opposing factions is crucial during training. To this end, we deploy distinct models for the werewolf and the "Good" factions. We find that optimizing Werewolves' speech instruction is much more challenging, as they need to mimic the "Good" faction's speech and master the art of disguise and deception. To mitigate this, we draw inspiration from Generative Adversarial Networks (Goodfellow et al., 2014) and adjust the training iterations, $n_{\mathrm{werewolf}} : n_{\mathrm{goods}} = 5 : 1$. To prevent actions and speech strategies from converging to a single pattern, we employ population-based training (Jaderberg et al., 2017) with a population size of $4$. We also introduce fictitious self-play (Heinrich et al., 2015), where in each game an average of 3 players employ the latest models, while the remaining 6 players use models randomly selected from the most recent 500 checkpoints. Further details on training pseudo-code, hyperparameters, reward shaping, and model structures are in Appendix F.

### 3.4 Presenter for Augmented Speech Generation

The generation of players' public speeches plays a crucial role in the Werewolf game, significantly impacting the outcome due to its strategic influence on actions and deductive reasoning of other players. The quality of the speech generation hinges on several critical aspects: (1) The strategy articulated within the speech should align with the player's role and the current state of the game. (2) Speeches need to adhere to the logical framework of the game, correlating with historical speeches and actions, making them sound and convincing. (3) Speeches are preferred to fit the stylistic environment of the Werewolf game. Detailed evaluation metrics can be found in Appendix F.2.

The Thinker module handles only the first aspect of speeches, providing a foundational strategic instruction for the Presenter, such as the Witch's decision to report the previous night's rescue, as shown in Figure 2. Subsequently, the Presenter leverages NLG capabilities of LLMs to craft a complete speech that incorporates the strategic instruction, relevant game state, and historical speeches. The template for the prompt is provided in Appendix F.7. Additionally, as with the Listener, we fine-tune the WereLLM to better align with human speech styles. The 260k speech-feature pairs are inverted such that the language feature **F** serves as the hindsight speech instruction **I**, and the actual speech **S** serves as output labels.

We have observed that LLMs often do not adhere to prompts, with even fine-tuned models sometimes producing hallucinations and inaccuracies. Taking inspiration from the Cicero (Bakhtin et al., 2022) approach, we introduce additional filtering steps. We use the Listener to perform further reasoning on the generated speech to produce a language feature, which we then compare for similarity to the

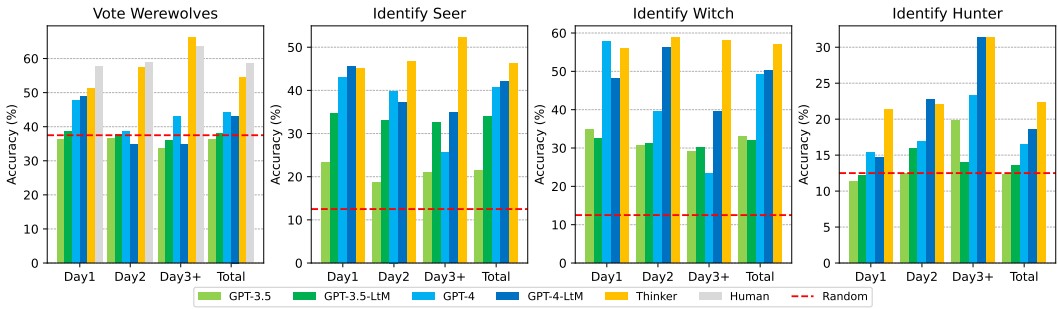

Figure 3: Voting and identity accuracy evaluating the deductive reasoning capabilities. The random baseline is calculated as the number of roles divided by the number of hidden players.

original speech instruction. For expressions detailing the speaker's own attributes, the filter demands an exact match. For expressions about others' attributes, the content indicated in the speech instruction must be consistent. For content not specified in the instruction, the filter allows the Presenter some flexibility, including minor hallucinations if they enhance the speech without detracting from its accuracy. The speech generation process repeats until it either meets the filter criteria or exceeds the maximum allowed attempts. Otherwise, a template-based speech is generated based on rules that consider the player's role, historical skills, and identity predictions.

# 4 EXPERIMENTS

We assess the performance of our framework by comparing it against several baselines and ablative variants. The models involved in the following experiments are as follows:

- **GPT-3.5/4**: GPT-3.5 and GPT-4 are directly applied to generate end-to-end action decisions and speeches. For GPT-3.5, we use the *gpt-35-turbo-16k* model, version *0613*. For GPT-4, we apply the *gpt-4* model, version *1106-Preview*. We prompt GPTs with game rules, explanations of typical game jargon, and comprehensive game information, including visible states, legal actions, and speech text converted by ASR. Detailed prompts are provided in Appendix F.7.

- **GPT-3.5/4-LtM**: This setting follows most aspects of the **GPT-3.5/4** configuration, except that we allow GPTs to first summarize each speech after being given 11 examples (as shown in Table 16), and then we let the GPTs generate actions and speeches based on the game state and all speech summaries. The Thinker module is not applied, thus no language features are retrieved.

- **GPT-3.5/4-T**: GPTs serve as the Listener and Presenter modules, while the Thinker module is integrated to generate actions and speech instructions. GPTs share the same prompts in setting **GPT-3.5/4-LtM**, except that additional speech instructions and identity predictions generated by the Thinker are added into the prompting of Presenter.

- **WereLLM-T**: We replace GPTs with the WereLLM in both the Listener and Presenter as an efficient practical solution, while the Thinker remains the same as in GPT-3.5/4-T. It is worth noting that our framework allows for the use of fine-tuned LLMs but does not require them.

## 4.1 DEDUCTIVE REASONING

We begin by evaluating the models' deductive reasoning capabilities. Based on the current game state, the historical actions and speeches, models are required to identify special roles (Seer, Witch, and Hunter) and vote for the most likely werewolf, from the perspective of villagers in the voting round each day. Given that villagers have a minimal amount of game information and must engage extensively in deductive reasoning within the game, this task represents a stringent test of the models' understanding and comprehension. From the *FanLang-9* dataset, we extract 300 games to serve as the test set, encompassing approximately 1200 evaluation instances. For the Thinker, we use its decision-making on actions for the werewolf voting task, and the identity model for identifying special roles. We assume that human players in the test set who are villagers would vote for the most likely werewolf. Thus, we list their voting choices as a reference, but their judgments regarding the identities of other players remain unknown.

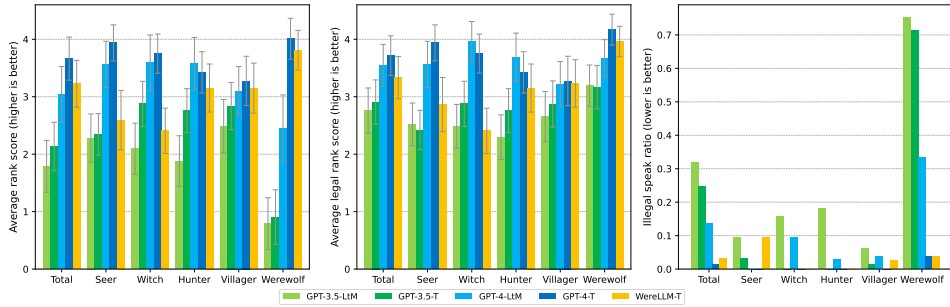

Figure 4: Human preference score for generated speeches grouped by roles. 10 evaluators are tasked with ranking the 2000 speeches following the criteria detailed in Appendix F.2.

Figure 3 presents the accuracy results. In terms of voting Werewolves, human players have the highest accuracy and the Thinker is closest to human players. The Thinker outperforms direct reasoning and prompting methods using GPTs in all the roles. LtM prompting enhances GPT-3.5's performance, especially in identifying the Seer, indicating advantages in processing complex and extensive speech contexts. However, the marginal gains of GPT-4-LtM over GPT-4 suggest that the latter's enhanced capability to process extensive texts reduces its reliance on speech summaries. In human gameplay, we observe that Seers and Witches often disclose their roles. This disclosure aids GPTs in outperforming random baselines, while Hunters and Werewolves typically conceal their roles, resulting in GPTs' performance aligning with random guessing. Notably, the accuracy of GPTs generally declines over successive days, except for the Hunter, whereas the Thinker's accuracy improves. This pattern suggests that although GPTs initially benefit from role disclosures on the first day, they may be hindered by the extensive speeches in subsequent days.

## 4.2 THINKER-AUGMENTED SPEECH GENERATION

We then investigate the capabilities of various models in generating speeches. Utilizing the same 300 complete games as discussed in Section 4.1, we extract 400 speech sessions that span a diverse range of roles, times of day, and speech types (first/second round speech, last words). Models are assigned the task of generating speeches based on the current game state and all players' historical speeches, with detailed prompts for GPTs available in Appendix F.7. Due to the demonstrated effectiveness of LtM prompting, subsequent experiments excluded GPTs that do not utilize LtM prompting. For GPTs-T and WereLLM-T settings, speech instructions are derived from the Thinker and incorporated into the prompts. We do not adopt the post filtering process for generated speeches in this experiment, which yielded approximately 2000 speeches for five models. To assess the quality of the single-shot generated speeches, we recruited 10 human evaluators, all well-versed in the Werewolf game. For each session, generated speeches are presented in a randomized order to ensure that evaluators are unaware of the model behind each speech. Evaluators are tasked with ranking the speeches and identifying any clear legal errors, following the criteria detailed in Appendix F.2.

The evaluation results are shown in Figure 4. In terms of total scores, models augmented with Thinker instructions outperformed their counterparts that relied solely on LtM prompting. Moreover, when augmented with the Thinker, the 6B WereLLM surpasses GPT-4-LtM. When examining scores for specific roles, the advantage of Thinker's contributions over GPT-3.5 appears somewhat marginal for the Seer, whose speeches are relatively straightforward, needing only to report inspections from the previous night. The assessment of villagers' speeches is inherently complex due to their limited available information, which is reflected in the minimal rank score differences observed among the models for this role. In contrast, differences on the rank score and illegal speak ratio are most obvious for Werewolves. This disparity stems mainly from the low legality of werewolf speeches, which often inadvertently reveal their identity. Remarkably, GPT-3.5 appears to struggle with adhering to instructions designed to avoid self-incrimination. In contrast, GPT-4 shows a more refined capability to disguise its identity, especially when augmented by the Thinker's strategic instructions. An example speech for the werewolf is presented in Figure 5.

Table 1: Online evaluation results showcasing the performance of 9 AIs using 5 different models and 3 combinations. Results are presented in the format: win rate | Behavior Score.

| Method | Total | Seer | Witch | Hunter | Villager | Werewolf |
|---|---|---|---|---|---|---|
| GPT-3.5-LtM | 36.7% \| −0.21 | 25.6% \| +0.16 | 23.1% \| −0.51 | 29.9% \| −0.21 | 30.8% \| −0.42 | 53.4% \| 0.00 |
| GPT-3.5-T | 47.4% \| **−0.05** | 38.3% \| +0.27 | **41.0%** \| **−0.14** | 36.4% \| **−0.12** | 33.8% \| −0.18 | 68.6% \| 0.00 |
| WereLLM-T | **50.3%** \| −0.06 | **38.8%** \| **+0.33** | 39.8% \| −0.18 | **37.0%** \| −0.29 | **39.1%** \| **−0.11** | 74.4% \| 0.00 |
| GPT-4-LtM | 37.9% \| **−0.01** | 21.9% \| +0.25 | 18.6% \| −0.25 | 19.4% \| **−0.06** | 20.3% \| **−0.00** | 73.6% \| 0.00 |
| GPT-4-T | 41.1% \| −0.02 | 20.4% \| +0.25 | 23.2% \| **−0.10** | **23.9%** \| −0.09 | 22.5% \| −0.09 | 78.4% \| 0.00 |
| WereLLM-T | **43.1%** \| −0.04 | **24.2%** \| **+0.27** | **24.6%** \| −0.15 | 23.4% \| −0.15 | **23.9%** \| −0.11 | **81.4%** \| 0.00 |
| GPT-3.5-LtM | 33.0% \| −0.22 | 14.4% \| +0.12 | 20.4% \| −0.46 | 20.7% \| −0.57 | 21.6% \| −0.33 | 57.0% \| 0.00 |
| GPT-3.5-T | 45.0% \| −0.07 | **33.6%** \| **+0.29** | 32.2% \| −0.13 | 30.4% \| −0.17 | 27.6% \| −0.20 | 75.8% \| 0.00 |
| GPT-4-LtM | 42.5% \| **−0.03** | 29.8% \| +0.27 | 22.2% \| −0.18 | 27.0% \| −0.20 | 28.7% \| **−0.04** | 71.9% \| 0.00 |
| GPT-4-T | **46.3%** \| −0.05 | 28.6% \| +0.28 | **34.5%** \| **−0.11** | **31.5%** \| **−0.08** | 28.0% \| −0.18 | **79.9%** \| 0.00 |
| WereLLM-T | 45.9% \| −0.06 | 29.1% \| +0.25 | 28.3% \| −0.16 | 29.2% \| −0.21 | **32.4%** \| −0.14 | 78.0% \| 0.00 |

## 4.3 ONLINE EVALUATION

Lastly, we conduct online evaluations to assess the overall performance of the models in a real-world gameplay setting. Given that Werewolf is a multiplayer game with imperfect information, the skill level of the participants can significantly influence the evaluation results. Therefore, we devise three combinations of models, with models being randomly and repeatedly selected to simulate 9-player games. We conduct approximately 600 rounds for each combination to ensure robust testing results. Given the inherent randomness of outcomes in the game, we also calculate the Behavior Score, a typical metric used in Werewolf competitions[4] to evaluate behavior of players, e.g., a Villager voting for a werewolf, a Hunter shooting a werewolf, a comprehensive breakdown is provided in Table 9.

The results summarized in Table 1 reveal that integrating the Thinker module significantly boosts the win rates of both GPT-3.5 and GPT-4 across all three model combinations. The performance of the WereLLM-T model closely aligns with that of GPT-4-T. In terms of Behavior Score, the Thinker contributes substantial improvements across all roles for GPT-3.5. For GPT-4, notable benefits are observed particularly for the Witch and Hunter roles. The Behavior Score metric assigns significant weight to the witch's poisoning and the hunter's shooting decisions, which correlates with the Thinker's ability to enhance werewolf detection and subsequently improve these scores. Another notable finding is that the combination involving GPT-4 and WereLLM-T models yields the highest win rate for Werewolves. This outcome primarily stems from the conservative nature of GPT-4-LtM in role identification, which leads it to be more cautious in voting and skill usage as the "Good" faction. In Appendix B.2, we also include an ablation of our framework with Cicero (Bakhtin et al., 2022) and LLM prompting-related approach Xu et al. (2023b), demonstrating that our method still maintains a significant advantage in terms of win rate.

Table 2: Online evaluation win rates with 1 human and 8AIs.

| Method | Total | Goods | Werewolves |
|---|---|---|---|
| GPT-4-T | 46.9% | 37.3% | 65.0% |
| WereLLM-T | 45.3% | 36.0% | 62.6% |
| Human | 40.5% | 35.3% | 59.4% |

Furthermore, to evaluate AI performance against human strategy, we incorporated 13 human players into the evaluation. We find that the issue of werewolf identity exposure, as illustrated in Figure 4, significantly impedes the game experience for human players. As a result, human evaluators play alongside four instances each of GPT-4-T and WereLLM-T models across 200 game rounds, and the post-filtering process for generated speeches is adopted in this setting. In Table 2, human players exhibit no significant win rate advantage, suggesting that the AI's speeches and actions do not exhibit exploitable weaknesses. Moreover, when compared with the results in Table 1, we observe a relative decrease in the Werewolves' win rate in games involving human players, highlighting the ongoing challenges related to identity concealment. Although AI-managed Werewolves might convincingly

---

[4]https://langrensha.163.com/20230313/31014_1077578.html

deceive other AI players, human players often find them suspicious. A typical example is that Werewolves tend to act in groups, such as unanimously voting for a certain player.

## 5 DISCUSSION AND LIMITATION

**Transfer to other tasks:** We use language features and speech instructions in our framework to integrate LLMs and external reasoning models. The communication format may not be directly transferable to other tasks or domains, with its effectiveness depending on the richness of these features and instructions. Future work aims to develop more generalized and flexible methods, such as using implicit hidden vectors in a data-driven manner, potentially offering better transferability at the expense of interpretability and controllability.

**Evaluation of 8 humans with 1 AI:** Our evaluations primarily involved games featuring either AI vs AI or one human player competing against multiple AIs. Evaluating an AI in a majority-human player setting presents challenges due to the highly interactive nature of the game and the variability in human players' speech strategies and behaviors.

**Interpretability:** While our framework improves the reasoning capabilities of LLMs, the reasoning processes in the Thinker module may not be easily interpretable to humans. We explicitly introduce the identity prediction task to reveal how the Thinker thinks of other players. Future work could explore methods for further improving the interpretability and transparency of our framework.

## 6 CONCLUSION

In this paper, we introduced a novel framework that integrates LLMs with an external Thinker module, aiming to enhance the reasoning capabilities of LLM-based agents. This approach is inspired by the dual-process theory and separates reasoning tasks into two systems: System-1, handled by LLMs, and System-2, handled by the Thinker. We showcased our approach in the context of the Werewolf game, a complex social deduction game requiring language processing, intuitive thinking, and strategic planning. Our results show that our framework can significantly improve the performance of LLMs and achieve better alignment with real-world scenarios and human preferences. Additionally, we fine-tune a 6B WereLLM to surpass GPT-4 when integrated with the Thinker. Furthermore, this paper contributes the largest dataset for social deduction games to date, aiming to accelerate advancements in this field.

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

**Day 1, first round speech sequence:**

8: 🧙 Seer > 9: 👩‍🌾 Villager > 1: 🧝 Witch > 2: 👩‍🌾 Villager > 3: 🐺 Werewolf > 4: 👩‍🌾 Villager > 5: 🏹 Hunter > 6: 🐺 Werewolf > 7: 🐺 Werewolf

**Speech History**

(8: 🧙 Seer): Hello everyone, I am Player 8 and my identity is a Seer. Last night I checked on Player 7, he was a werewolf.
(9: 👩‍🌾 Villager): …
(1: 🧝 Witch): I am the Witch saving myself. I suspect that Player 6 is a werewolf and I will poison him. Player 9 sounds like a good man.
(2: 👩‍🌾 Villager): I am a Villager, let's listen to Player 7's statement.

**Speech without Thinker Instruction (GPT-4)**

(3: 🐺 Werewolf): Fellow villagers, I am Player 3. Based on the previous speech, I think Player 9 is a good guy because he said nothing but external information, which is an opportunity **for us werewolves** ❌. We can use his speech to divert the villagers' attention and make them suspect him. As for Player 6, I don't have enough information to make a judgment. Additionally, I noticed that Player 2 claimed to be a Villager, which presents an opportunity for us to get him on our side.

**Thinker-Augmented Speech (GPT-4)**

**Speech instruction:**

- "identities": {"seer": [3], "good": [9]},
- "actions": {"check": [[3,9]], "vote out": [[3,8]]}

(3: 🐺 Werewolf): Hello everyone, **I am the Seer** ✅. Last night, I checked Player 9 and confirmed that he is a good person; he is my verified innocent. I will protect him to the end. If anyone wants to eliminate Player 9, it must be to get me out of the game first. Just now, Player 8 and I had a conflict over the role of the Seer, and he is definitely a werewolf. **Therefore Player 7 might be a good man**. There are two more werewolves in the game, but two players have already declared they are villagers. There aren't enough roles for villagers left. I will listen carefully to the subsequent players' statements. Everyone, let's vote out Player 8 this round.

Figure 5: An example comparison of speeches with and without strategic instruction.

# A    DESIGN PRINCIPLES

Regarding related works in Figure 1, we detail the evolving process of our framework as follows.

## A.1    MOTIVATION

In the game of werewolf, there is a significant gap between what a player says and what the player is actually thinking. Consider the scenario depicted in Figure 5, where Player 3, a werewolf, publicly states:

*"I am the Seer, and I have checked Player 9, who is a good person. I suspect that Player 8 is a werewolf."*

While the surface meaning of this speech (*System-1*) is straightforward, the internal thought process of Player 3 (*System-2*) might be as follows:

*"Players 6 and 7 are my fellow Werewolves (as per the game rules, Werewolves know each other's identities), and Player 8 claims to be the Seer and has accused Player 7, who is on my team. Therefore, Player 8 is likely the real Seer. By also pretending to be the Seer and verifying Player 9 as a good role, I can create a conflict with Player 8 in the eyes of the villagers."*

## A.2    LLM PROMPTING METHODS

We identified several shortcomings when examining the performance of LLM with typical prompt or mechanism engineering methods. The shortcomings can be categorized into two main areas:

**Over-trust**: LLMs exhibited a tendency to over-trust other players' self-declared identities, particularly when players claimed to be Seer or Witch roles. Furthermore, when the LLM assumed the role of a Werewolf itself, it was prone to inadvertently exposing its own identity, which is demonstrated in Figure 4 and Figure 5.

**Strategic Deficiencies**: LLMs showed a lack of familiarity with the common strategies employed in the Werewolf game. For instance, they failed to grasp tactics such as Werewolves pretending to be Seers to mislead other players, Werewolves accusing their teammates to gain the trust of the "Good" players, or Villagers pretending to be Seers to protect the real Seer from being killed, etc. These are conventional tactics used by experienced human players to navigate the complex social dynamics of Werewolf, which involve deception, trust, and betrayal.

To delve deep into the reasoning process of LLMs, we dissected the process from listening to speaking in the game into four stages, as shown in Figure 1 and investigate issues one by one:

(1) **NLU**: It is assigned as the Listener's goal in Figure 2, is to interpret speeches and extract their explicit meanings. LLMs show proficiency in this area.

(2) **Deductive reasoning**: LLMs underperform in role identification, often over-trust other players' self-declared identities, as tested in Section 4.1. Then the deductive reasoning is limited to information extraction.

(3) **Speech strategic planning**: LLMs struggle to outline a comprehensive speech plan, especially when assuming the role of a Werewolf. They frequently risk exposing themselves or their allies (see Figure 5), lacking an understanding of conventional Werewolf game speech strategies.

(4) **NLG**: Although LLMs are unfamiliar with conventional speech strategies, we find that they can generate coherent and convincing speeches once prompted with basic instructions, such as "You should pretend to be the Seer and accuse Player 3 of being a werewolf".

### A.3 TRANSITION TO THE THINKER MODULE

The primary reason for the above shortcomings is that LLMs are not trained on Werewolf-specific knowledge corpus and data. Although it is possible to prompt LLMs with common game terminologies through in-context learning, strategic experiences are challenging to encapsulate in text prompts. To address the deficiencies in deductive reasoning and speech strategic planning, we consider developing a trainable Thinker model to handle these aspects separately from the LLMs. The Thinker module is optimized through imitation learning and reinforcement learning, using human game data as a foundation. It is designed to complement the LLMs, then the latter are responsible for intuitive, domain-agnostic *System-1* reasoning tasks.

### A.4 COMPARISON WITH CICERO

In brief, the differences between our approach and Cicero are as follows:

**Different Roles for NLU and NLG**: In Cicero's approach, both NLU and NLG involve a high-level logical reasoning process: NLU directly outputs action predictions, which is actually a complex reasoning process that goes beyond natural language processing. Similarly, NLG takes intended actions as control signals, but it still requires a comprehensive consideration of the game state, historical speeches, and higher-level reasoning to generate reasonable dialogue/speech that matches the intended action. In contrast, in our Werewolf game approach, the Listener (NLU) is only responsible for extracting key information from speeches and does not infer the truthfulness of the speeches or the underlying intentions. Similarly, NLG expands speech instructions, which are outlines of speeches, into full statements in context, requiring less domain-specific reasoning.

**The Connection between LLMs and Policy**: In Cicero's approach, the connection between LLMs and policy is made only through action prediction and intended action, which is non-language-based. In the Werewolf game scenario, we found that using actions alone is not sufficient, as the Listener causes significant information loss. Due to the complexity of Werewolf speeches, intended actions also struggle to describe and control speech generation. This leads to a noticeable disadvantage for Cicero's approach in the ablation study presented in Table 3 and Table 4. To address this, we propose a language-based feature and speech instruction that include complex verbal information, which can effectively summarize player speeches and control the speech generation process.

**Different Training Modes**: Due to Cicero's method involving NLU and NLG in task-specific high-level reasoning processes, it is necessary to fine-tune both NLU and NLG. In our approach, by defining explicit language-based connections and isolating domain-specific complex reasoning from LLMs with the Thinker, we can avoid the fine-tuning of NLU and NLG.

### A.5 COMPARISON WITH OTHER DUAL-SYSTEM METHODS

Our work mainly differs in that we utilize an RL model for system-2 reasoning (**deductive reasoning** and **strategic planning** in the Werewolf game), while the related work primarily relies on LLMs for complex, domain-specific system-2 reasoning, as shown in Figure 6. We believe that employing an RL policy model for system-2 reasoning is a viable choice in complex decision scenarios (for example, GPT4 + AlphaGo) and specific domain contexts where LLM pre-training does not include

Table 3: Accuracy of predicting future actions.

| Time | Total | Night skills | | | Day actions | |
|------|-------|------------|-------|------|--------|------|
| | | Werewolves | Witch | Seer | Hunter | Vote |
| Day1 | 37.0% [422/1142] | 13.3% [40/300] | 97.0% [97/100] | 12.0% [12/100] | 0.0% [0/4] | 42.8% [273/638] |
| Day2 | 30.3% [268/884] | 17.0% [51/300] | 20.6% [20/97] | 18.4% [14/76] | 10.0% [1/10] | 45.4% [182/401] |
| Day3+ | 36.6% [128/350] | 34.4% [67/195] | 30.0% [3/10] | 22.7% [5/22] | 33.3% [1/3] | 43.3% [52/120] |

domain knowledge. In such cases, LLMs primarily function in the role of NLP, corresponding to the Listener and Presenter in our text.

| Method | System1 | | | System2 | | |
|--------|---------|----------|-----------|---------|----------|-----------|
| | Model | Function | Alignment | Model | Function | Alignment |
| Swiftsage | Small LM | Generate actions | Finetune | LLM (GPT4) | subgoal planning and grounding | Prompting |
| DECKARD | Policy model | Execute subgoals, action and control | RL | LLM | Decompose and subgoals | Prompting |
| Ours | LLM | NLP related, explainability | Prompting or SFT | Policy model | deductive reasoning and strategic planning | RL |

Figure 6: Comparison of system components and functionalities.

# B ADDITIONAL RESULTS AND ABLATION STUDIES

## B.1 PREDICTING ACTION AS LANGUAGE FEATURES

We study the approach used by Cicero (Bakhtin et al., 2022), utilizing the prediction of players' future actions as a feature representation of speeches and as a control variable for the speech generation. Aside from the example illustrated in Figure 2, we additionally conduct experiments by feeding the model with complete game states and historical speeches to predict players' future actions. We fine-tune the WereLLM model using data from the *FanLang-9* dataset and then test the action prediction accuracy on a set of 100 test games.

The results are shown in Table 3. Overall, the action prediction accuracies for three days do not exceed 40%. Notably, the Witch conventionally saves the player killed by Werewolves on the first day, resulting in a high accuracy. One point of particular interest is the accuracy of voting predictions, which consistently remained just over 40% as the days progressed. In the game of Werewolf, the speaking order plays a crucial role; players who speak earlier often mention multiple potential voting targets. By listening to subsequent speeches, players can make informed decisions or adjustments regarding their final vote. This aspect of the game dynamics makes the implementation of Cicero's method challenging in the context of Werewolf.

## B.2 COMPARISON WITH OTHER APPROACHES

In this section, we compare the performance of our proposed method, a Cicero-like baseline variant, and the approach described in Xu et al. (2023b). To ensure a rigorous experimental comparison, we adapted the implementations of the comparative methods to account for differences in implementation details, thereby enhancing the persuasiveness of our results. Below we outline the configurations for each method:

**Our Method**: We employ the **GPT-4-T** setting, wherein the Listener and Presenter modules utilize GPT-4, and the Thinker module is powered by the RL-optimized model.

**Variant of Cicero**: For this baseline, we reduce the language feature and speech instruction dimensions to a single dimension, representing the future action of a speaking player. As experimental findings in Appendix B.1 indicated that fine-tuning WereLLM yielded low action prediction accuracy, we directly use GPT-4 to generate language features and speech instructions in the Listener and Presenter. The Thinker module employs an RL model for training, with its language feature and speech instruction also condensed to one dimension. All other configurations are consistent with GPT-4-T.

**Variant of Xu et al. (2023b)**: Diverging from the original implementation, we modify the approach to have GPT-4 generate three speech instruction candidates instead of directly producing speak candidates. The Thinker then selects one speech instruction, which is subsequently used by the GPT-4 Presenter to generate speech. Due to the discrepancy between LLM inference and Thinker RL sampling speeds, the Thinker is restricted to using offline RL. For offline RL data construction, we extracted 1000 game sessions from the *FanLang-9* dataset. For each instance of speaking, we allow GPT to generate five speech instruction candidates. During offline RL training, we randomly selected two of the five GPT-generated candidates and combined them with the human speech instruction to form three speech instruction candidates, yielding 10 possibilities for data augmentation. The Thinker makes its selection, with its inputs including the game state, language features as in GPT-4-T, and the three speech instruction candidates. The actual selection for BC is the human speech instruction.

To summarize, the primary distinction between GPT-4-T and the Cicero variant lies in the modification of the dimensions and meanings for language feature and speech instruction. And the Thinker in the variant of Xu et al. (2023b) no longer generates speech instructions; instead, it directly selects from generated candidates. The evaluation results are shown in Table 4. Our GPT-4-T method surpasses the variant of Xu et al. (2023b) in performance, and significantly outperforms the Cicero variant, highlighting the advantages of external Thinker module in terms of reasoning and strategic communication within the Werewolf game.

Table 4: Win rate comparison of our method with other approaches.

| Method | Total | Goods | Werewolves |
| --- | --- | --- | --- |
| Variant of Cicero (Bakhtin et al., 2022) | 34.4% | 28.5% | 47.9% |
| Variant of Xu et al. (2023b) | 47.8% | 37.4% | 67.7% |
| Ours (GPT-4-T) | **53.5**% | **41.6**% | **75.2**% |

B.3    COMPARISON WITH OTHER PROMPTING APPROACHES

We compare our method with more complex prompting approaches. The experiment includes three configurations:

- **GPT-4-LtM**: The same as in the main text, except that we switch to the *gpt4-turbo-2024-04-09* model.

- **GPT-4-T**: The same as in the main text, except that we switch to the *gpt4-turbo-2024-04-09* model. It is worth noting that the LtM prompting is applied in this setting.

- **GPT-4-LtM-ReAct**: The same as the **GPT-4-LtM** configuration, except that we additionally apply ReAct (Yao et al., 2022) prompting. Each time it is the player's turn, we allow the GPT to analyze and guess the role of other players and their future skills and voting decisions, and buffer all the historical guessing and thinking as additional information in the prompts.

We conduct 500 rounds for the combination to ensure robust testing results. The results are shown in Table 5. As can be seen, compared to GPT-4-LtM-ReAct and GPT-4-LtM, the additional, more complex ReAct prompting provides limited improvement (1%) in GPT's capabilities. GPT-4-T shows a significant improvement over GPT-4-LtM-ReAct, even though it only uses LtM prompting. This indicates that the Thinker module plays a key role in the reasoning of the Werewolf game.

Table 5: Win rate comparison of our method with other prompting approaches.

| Method | Total | Goods | Werewolves |
|---|---|---|---|
| GPT-4-LtM | 41.2% | 28.2% | 68.8% |
| GPT-4-LtM-ReAct | 42.3% | 28.5% | 69.5% |
| GPT-4-T | **45.7%** | **30.9%** | **74.0%** |

### B.4 TRANSFER TO 6-PLAYER WEREWOLF GAME

To demonstrate the generalizability of our framework, we transfer the Thinker module trained on a 9-player Werewolf game to a 6-player Werewolf game. The 6-player Werewolf game consists of 2 villagers, 2 werewolves, a Seer, and a Savior. Each night, the Savior can choose to protect a player from being killed by the werewolves, but the protection cannot be given to the same player on two consecutive nights. The Savior in the 6-player game is similar to the Witch in the 9-player game, except without the poison and with an unlimited supply of antidotes.

The Thinker module is initially trained by RL and BC in the 9-player game, then fine-tuned in the 6-player game mode with RL. Since speeches in the *FanLang-9* dataset (9-player) might violate the rules of the 6-player game, e.g., "Player 9 is Seer", "Players 2, 5, 6 might be werewolves", we cannot easily fine-tune the WereLLM and transfer it to the 6-player game. Therefore, we test the combination of GPT-4, GPT-4-LtM, and GPT-4-T. The WereLLM-T setting is not included in this ablation. The details of these three models are the same as in the main text, except that we switch the GPT behind these three settings to the *gpt4-turbo-2024-04-09* model.

The results are shown in Table 6. The experimental conclusions are quite similar to those in Table 1. Compared to GPT-4-LtM and GPT-4, the LtM prompting provides limited improvement in GPT's capabilities. GPT-4-T shows significant improvement over GPT-4-LtM.

Table 6: Win rate of Thinker module transferred in the 6-player Werewolf game.

| Method | Total | Goods | Werewolves |
|---|---|---|---|
| GPT-4 | 48.7% | 50.4% | 45.3% |
| GPT-4-LtM | 50.1% | 51.4% | 47.3% |
| GPT-4-T | **53.1%** | **53.7%** | **51.8%** |

### B.5 TRAINING CURVE

The population-based RL training of different agents is illustrated in Figure 7.

## C GAME RULES

We follow the 9-player standard mode Werewolf game rules on the Fanlang platform. The rules are outlined as follows.

### C.1 OBJECTIVES

The game is divided into two factions: the "Good" faction, which includes Villagers and special roles, and the "Werewolf" faction. Additionally, there is a Moderator who is responsible for managing the game and ensuring the rules are followed. The goal for the "Good" faction is to identify and eliminate all Werewolves, while the Werewolves aim to kill or exile all Villagers and special roles. The game ends when any of the following conditions are met:

• All Villagers are out of the game (Werewolves win)
• All special roles are out of the game (Werewolves win)
• All Werewolves are out of the game ("Good" faction win)

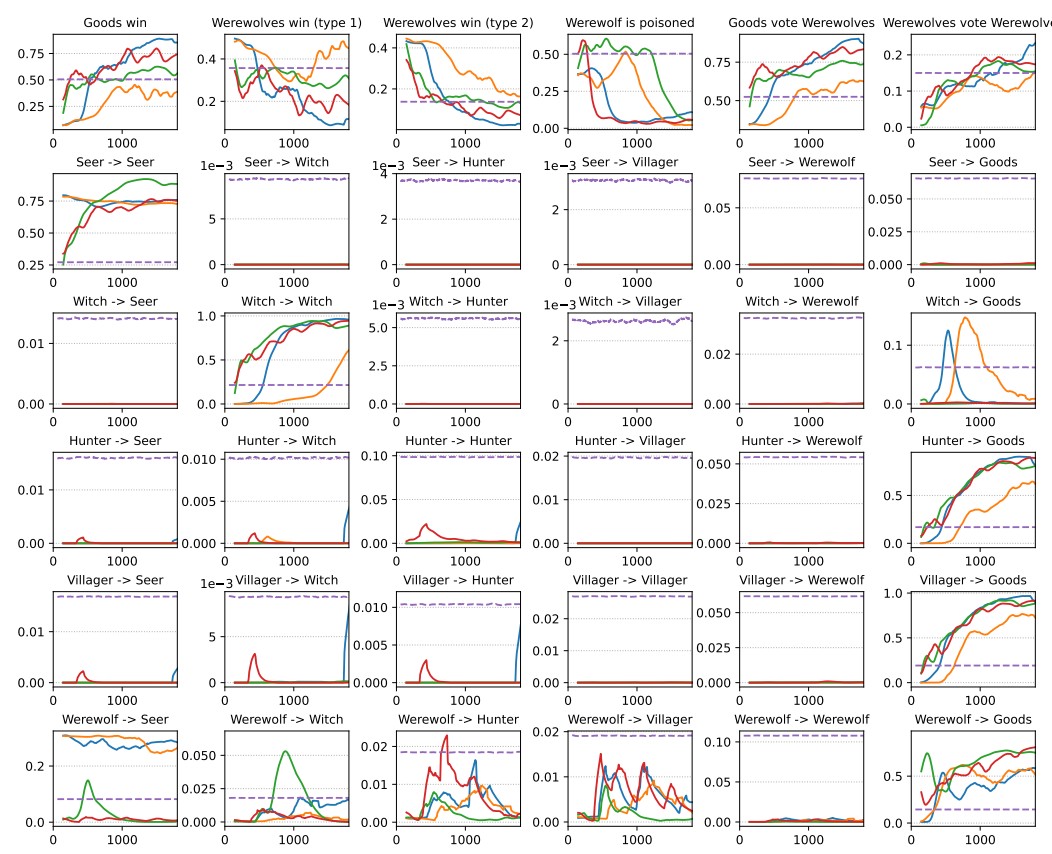

Figure 7: Detailed training curves for different agents during RL training. The x-axis represents the training steps (k), and the y-axis represents the probability. The horizontal line in each subplot corresponds to the probability observed in human data. "Werewolf -> Seer" represents that a Werewolf claims to be the Seer in the speech.

## C.2 ROLES

The game comprises 3 Villagers, 3 Werewolves, and 3 special roles (Seer, Witch, and Hunter). The identities of the players are hidden from each other, even after being eliminated from the game.

**Werewolves**: Werewolves are aware of each other's identities. At night, they decide to kill a living player, which may include one of their own. The majority of the Werewolves' choice will be the final kill target. If there is a tie, a random player in the tie is killed. Werewolves can commit suicide during the speech sessions, which will reveal their identity, and the game immediately proceeds to the night phase, skipping the remaining daytime processes such as speeches and voting.

**Villagers**: Villagers have no special abilities. They must determine other players' identities based on their speeches and vote to exile potential Werewolves.

**Seer**: The Seer can verify a player's faction each night (either Werewolf or "Good"), but cannot discern their specific role. The Seer cannot verify himself or any player who has already been verified.

**Witch**: The Witch possesses an antidote and a poison. The antidote can save a player killed by Werewolves at night, and the poison can kill a player. The Witch cannot use both potions in the same night and can only save herself on the first night.

**Hunter**: When the Hunter is killed by Werewolves at night or voted out during the day, he can shoot a player. However, the Hunter cannot use his ability when poisoned by the Witch.

## C.3 GAME TASK FLOW

The game proceeds in a night-day cycle until the victory conditions are met.

**The night tasks flow**:

(1) Werewolves decide to kill a player. In our simulation of the game environment, we have **simplified the discussion into a three-round voting process**. During voting, werewolf players can see their teammates' previous votes.
(2) The Witch uses her ability.
(3) The Seer uses his ability.

**The daytime tasks flow**:

(1) The Moderator announces the deaths from last night but does not reveal the causes of death.
(2) Deceased players give their last words (only for the first day).
(3) If deceased players have additional abilities, they may choose to use them.
(4) First round of speeches. The speech sequence is determined by the following rules: (a) if no player died last night, randomly select an initial speaker and randomly decide a clockwise or counterclockwise speaking order. (b) A deceased player is randomly selected, and the speaking order starts clockwise or counterclockwise from him. Players cannot interrupt others during their speeches.
(5) First round of voting. Each player votes for a single player to exile from the game. Other players' voting choices remain hidden until the voting session ends.
(6) Second round of speeches. If there is a tie in the first round of voting, the tied players give their second speeches; otherwise, the process moves on to task (8) The first speaker, selected randomly from the tied players, initiates the sequence, which could proceed either clockwise or counterclockwise.
(7) Second round of voting. If there is still a tie after the second vote, the game moves on to the next night, and no player is exiled.
(8) The exiled player gives his last words.
(9) If exiled players have additional abilities, they may choose to use them.

## D ANALYSIS OF THE *FanLang-9* DATASET

The *FanLang-9* dataset consists of 18 800 recordings and 260K speech instances, with an average speech length of 500 characters. Specifically, the following characteristics underscore the unique nature of the dataset:

### D.1 SPEECH DURATION AND LENGTH

Figure 8 (a) demonstrates significant variations in speech duration among different roles, with an average of approximately 90 seconds each. The Seer's inspection information at night forms the core and fundamental logical basis of the game. Therefore, it is the Seer's duty to share inspection information, provide persuasive speeches, and lead discussions during the speech phase, resulting in the longest duration among all roles. Besides, Werewolves and Villagers need to convincingly identify themselves and predict the roles of other players, necessitating detailed and logical analysis.

In Figure 8 (b), the dataset shows the shortest token length among Werewolves, which is not correlated with their speaking time. This suggests that Werewolves' speeches are relatively concise, which may stem from the complexity of deception that requires more time to strategize. We further illustrate the distribution of token length in a single speech in Figure 9.

### D.2 TOKENIZATION AND CATEGORIZATION OF SPEECHES

The reasoning result of a speech produced by the Listener is formatted in JSON style, containing pairs of player IDs with their attributes. The result typically includes phrases and word groups containing multiple attributes, probabilities, and irrelevant information, e.g., "seems to be a werewolf: [3, 6]", "cannot hear clearly: [8]". We then tokenize and categorize the result into related identities and

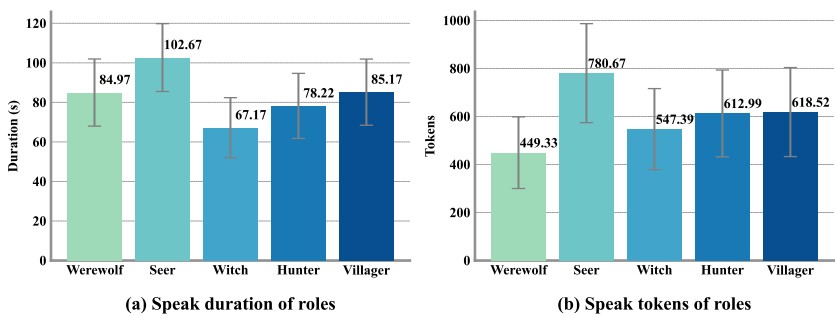

(a) Speak duration of roles    (b) Speak tokens of roles

Figure 8: Speech duration and token length categorized by roles in the *FanLang-9* dataset.

actions, along with their probabilities, as shown in Table 8. The final language features account for 96.09% of the *FanLang-9* dataset, capturing the majority of the information expressed by speakers in the Werewolf game.

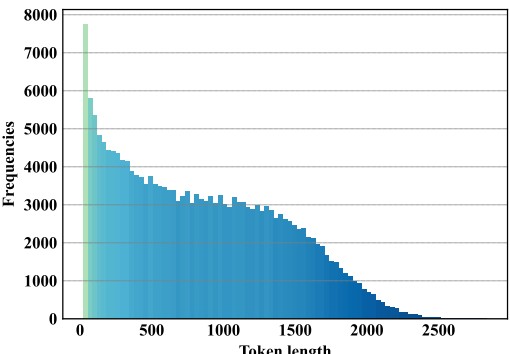

Figure 9: Distribution of speech token length.

### D.3 VOTING PREFERENCE

We analyze how human players tend to vote in the perspective of different roles in Figure 10 (a). As for voting Werewolves, the Seer has the highest accuracy of identifying Werewolves due to his inspection ability, while Werewolves vote for their teammates with a probability of 15.7%, aiming to disguise themselves as the "Good" faction. The other roles have a 50% chance of voting for Werewolves, since they lack additional information beyond the game state and historical speeches. As for voting from Werewolves, the most prioritized target are the Villagers (28.6%), since they have the least amount of information and are easier to be incriminated as Werewolves. The second prioritized target is the Seer (28.1%), since the Seer can inspect players' identities, it is crucial to remove him out of the game as soon as possible.

### D.4 FINAL STATE OF THE ROLES

In Figure 10 (b), we present the final states of roles in the end of the game, categorized as *Survived*, *Shot* by the Hunter, *Poisoned* by the Witch, *Killed* by Werewolves, *Exiled* after the Voting stage, and Werewolves committed *Suicide*. Notably, the Witch has the highest likelihood of being killed by Werewolves at night (55.3%), with the Seer following at 32.5%. Werewolves commit suicide with a probability of 17.3%, and are killed by their teammates at night with a probability of 2.5%. During the daytime voting, Werewolves are the most frequently exiled role, indicating their challenges in providing deceptive statements, while the Witch has the lowest probability, reflecting their effectiveness in gaining trust through speeches.

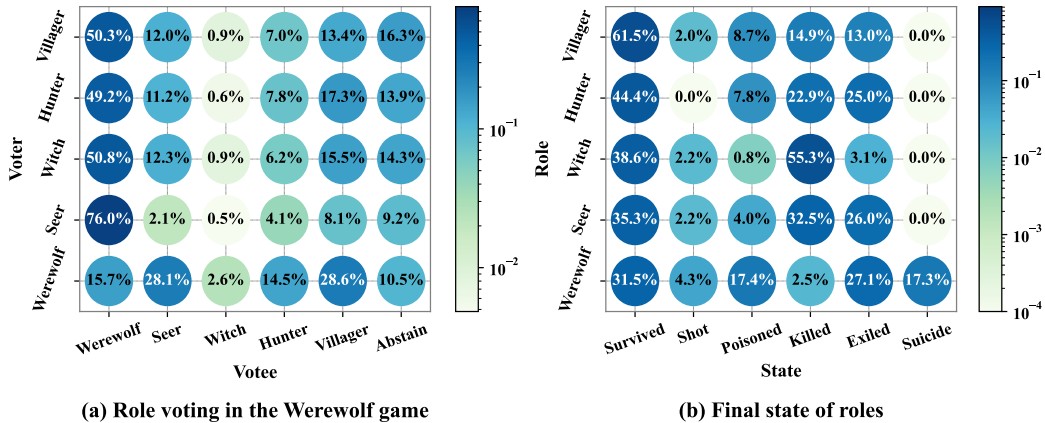

(a) Role voting in the Werewolf game  (b) Final state of roles

Figure 10: (a) Voting probability distributions for players with different identities across all voting sessions; (b) Final survival status and causes of death probabilities for players at the end of the game.

## D.5  WIN RATE

Table 7 illustrates that in human gameplay, the win rates for the Good and Werewolf factions are closely matched.

Table 7: Win rate in the *FanLang-9* dataset.

| Camp | Win number | Win rate |
|------|-----------|---------|
| Goods | 9293 | 49.31% |
| Werewelf | 9554 | 50.69% |

Table 8: Tokenization and categorization of speeches on the *FanLang-9* dataset.

| Tokenized attributes | Is | Might be | Is not | Might not be | Is not sure | Ratio | Accumulation |
|---|---|---|---|---|---|---|---|
| Werewolf | 178 423 | 27 297 | 516 | 313 | 15 | 26.55% | 26.55% |
| Good (the good faction) | 83 071 | 622 | 85 | 73 | 10 | 10.77% | 37.32% |
| Vote | 68 853 | 87 | 81 | 1 | 3 | 8.87% | 46.19% |
| Seer | 60 339 | 114 | 111 | 321 | 8 | 7.82% | 54.01% |
| Witch | 35 408 | 42 | 29 | 8 | 3 | 4.56% | 58.57% |
| Gold Water (checked Good) | 34 727 | 8 | 8 | 1 | / | 4.46% | 63.03% |
| Check (Seer's inspection) | 26 027 | 17 | 17 | / | / | 3.35% | 66.38% |
| Poison | 21 897 | 82 | 9 | 1 | / | 2.83% | 69.21% |
| Villager | 21 611 | 28 | 19 | 10 | 1 | 2.78% | 71.99% |
| Werewolves' target | 19 481 | 17 | 12 | / | 1 | 2.51% | 74.50% |
| Hunter | 17 603 | 26 | 70 | 5 | 2 | 2.28% | 76.78% |
| Silver Water (saved) | 14 016 | 3 | 5 | 1 | 2 | 1.80% | 78.58% |
| Suicide | 3826 | 4 | 1 | / | 1 | 0.49% | 79.07% |
| Uncertain Identity | / | / | / | / | 2937 | 0.38% | 79.45% |
| Shoot | 1100 | 2 | 2 | / | / | 0.14% | 79.59% |
| Save (by the Witch) | 1065 | / | / | / | / | 0.14% | 79.73% |
| Abstain voting | 683 | 3 | 1 | / | / | 0.09% | 79.82% |
| Special Role | 273 | 4 | / | / | / | 0.04% | 79.86% |
| Irrelevant Information | 126 279 | / | / | / | / | 16.23% | 96.09% |
| Unprocessed | 30 476 | / | / | / | / | 3.91% | 100.00% |

## E   ETHICAL CONSIDERATIONS

With the integration of LLMs into complex reasoning tasks, as demonstrated in social deduction games like Werewolf, we are witnessing the emergence of AI agents. These agents not only mimic human-like reasoning but also engage in communications that could inherently be considered deceptive. While these developments showcase the potential of AI to understand and navigate intricate human interactions, they also raise important ethical and societal considerations that must be addressed. To address these ethical and societal challenges, we propose several mitigation strategies:

**Transparent Communication and Monitoring**: Our framework ensures transparency through explicit structured information at every stage of the AI's decision-making process, from listening and reasoning to speech generation. To enhance this transparency, we propose implementing real-time transparency logs that capture and display the reasoning paths, identity predictions, and speech instructions generated by the AI. By having a complete audit trail, we can monitor the AI's decision processes, ensure adherence to ethical guidelines, and trace any unintended actions back to their source.

**Control and Filtering Mechanisms**: Our speech instructions are enriched with contextual information specific to the Werewolf game, allowing for robust control over the fine-tuned LLM. To further mitigate potential negative impacts, we propose implementing dynamic contextual guardrails. These guardrails will utilize our existing filtering mechanism (as outlined in Section 3.4) to not only match generated speech with instructions but also to check against a set of ethical and societal norms. If the AI's output is flagged as potentially harmful or deceptive beyond the game's scope, it will be withheld and replaced with a template response. This additional layer of control will act as a safeguard against the misuse of AI in generating deceptive or manipulative content outside the intended gaming environment.

## F   IMPLEMENTATION DETAILS

### F.1   DETAILS FOR HUMAN EVALUATORS

We recruited 13 human players to participate in the online evaluation of 1 human and 8AIs in Table 2. And 10 of them further participated in the evaluation of speech generation in Figure 4. The participants are selected from a board game association consisting of over 60 people, and the association regularly hosts offline Werewolf games. The evaluators are not paid for the evaluation and are required to be familiar with the rule of 9-player Werewolf games and have participated in at least 20 online/offline Werewolf games. Due to the multiple different versions of the Werewolf game, before the evaluation, we provided all evaluators with a detailed demonstration based on data analysis from 18 000 *FanLang-9* game sessions, including the probabilities of different strategies for different roles, and required them to watch at least 10 game recordings from *FanLang-9*.

### F.2   CRITERIA FOR EVALUATING SPEECH GENERATION

The human evaluation requirements for speech generation are as follows.

**Legality:** Absence of obvious logical errors and statements that conflict with the game rules, such as:

- "I am a Werewolf."
- "I am the Seer, and I poisoned Player 5 last night."
- "Player 3 is a good person; I suggest voting for him."
- "I suggest voting for myself."
- "Player 8 is a Werewolf, he was voted out and took Player 6." (Incorrect: Player 8, as the hunter, publicly shot Player 6).
- "I suggest voting for Player 8." (Incorrect: Player 8 has already been voted out).

**Reasonableness:** of the speeches, such as:

- The Seer correctly reports his inspection last night.

- Werewolves reasonably disguise their identities by employing various strategies, such as pretending to be the Seer, making aggressive claims, and betraying their teammates.
- Villagers make reasonable guesses about the Good faction and Werewolves.
- Note: the correctness of guessing other players' identities is not part of the evaluation criteria.

**Other:** factors unrelated to key information:

- Language style, colloquial expression, game jargon.
- Presence of verbose or redundant statements, such as greetings or defending the village community.

The evaluation criteria are in descending order of priority. For example, if model A has no obvious logical errors but its speech is not very reasonable, and model B has obvious logical errors, then A is better than B. When ranking the five samples, mark any with obvious logical errors as $-1$; these do not require further ranking. For example, if models A and B have obvious errors, the annotation result could be: $A : -1, B : -1, C : 1, D : 2, E : 3$, where 1 represents the best and 5 represents the worst. Apart from marking illegal statements as $-1$, tied rankings are not allowed.

Table 9: Behavior scores applied in the 9-player werewolf game.

| Role | Description | Score |
|---|---|---|
| Seer | If a werewolf is exiled in the first day | $+0.5$ |
| | For giving up the inspection at night | $-0.5$ |
| Witch | For poisoning a werewolf | $+1.0$ |
| | For poisoning a good player | $-1.0$ |
| Hunter | For shooting a werewolf | $+1.0$ |
| | For shooting a good player | $-1.0$ |
| Good roles except the Seer | For voting for a werewolf | $+0.5$ |
| | For voting for a good player | $-0.5$ |

### F.3 MODEL STRUCTURE OF THE THINKER

The Thinker network's architecture is designed to capture the intricacies of gameplay from the current player's perspective. It encompasses speeches, actions, and game status information for all nine involved players, including the player itself. We employ a shared-parameter feature encoding network to process the data for each of the nine players individually.

For the $i$-th player, up to 10 language features $\mathbf{F}$ are stored. These language features are enriched with headers indicating the time-tag, type, and order of the speeches. Subsequently, these annotated language features are processed through another shared-parameter speech feature encoding network, which consists of a three-layer multilayer perceptron (MLP) network (181-256-256). After processing the ten pieces of features, a *reduce_mean* operation is applied to the outputs to synthesize the overall speech embedding for the player $e_i^{\text{speech}}$. This synthesized speech embedding is then combined with additional game state information such as the player's actions, status, and other relevant data. The aggregated data is fed through a feature encoding network (again, a three-layer MLP of 1019-512-512) to generate the feature embedding for the $i$-th player $e_i$.

In the final step, the feature embeddings of all nine players $e_1, e_2, ..., e_9$ are subjected to a *reduce_mean* operation to create a collective feature encoding. This comprehensive encoding is then passed through an all-player feature encoding network (a three-layer MLP of 523-512-512) to construct the corresponding action decision, identity prediction headers, as well as speech instructions.

### F.4 REWARD SHAPING

Inspired by the Behavior Score concept, we have devised a reward shaping strategy for the Thinker in the reinforcement learning to circumvent illegal actions and speech that may arise during unfettered

exploration within the AI Werewolf game. The specifics of this mechanism are outlined in Table 10. It encompasses several key areas:

- Game result reward: The AI receives a reward based on the game's outcome (win or loss) and the survival duration.
- Action reward: for taking actions that are deemed appropriate and effective within the context of the game.
- Speech reward: incentivizing the AI to engage in communication beneficial to its goals, such as persuading other players or disseminating useful information.
- Action-Speech consistency reward: awarded for coherence between the AI's declared intentions in speech and its subsequent actions.
- Cognitive reward for Werewolves: Central to the training of a Werewolf AI is the ability to masquerade as a member of the "Good" faction. To enhance this capability, we provide a reward based on the change in identity prediction from the perspective of the "Good" players. The better a Werewolf AI can deceive the "Good" faction about its true identity, the larger the reward it receives.

Table 10: Reward shaping in the RL training of the Thinker.

| Description | Reward |
| --- | --- |
| # Game reward | |
| the Good faction win, Werewolves get | $-4$ |
| the Good faction win, Villagers and special roles get | $+2$ |
| Werewolves win, Werewolves get | $+4$ |
| Werewolves win, Villagers and special roles get | $-2$ |
| Any player survives for a new day | $+1$ |
| # Action reward | |
| the Goods vote for a Werewolf | $+2$ |
| the Goods vote for a Good role | $-2$ |
| the Witch poisons a Werewolf | $+2$ |
| the Witch poisons a Good role | $-4$ |
| the Hunter shoots a Werewolf | $+2$ |
| the Hunter shoots a Good role | $-4$ |
| # Speak reward | |
| the Seer claims his identity | $+2$ |
| the Witch claims his identity | $+1$ |
| the Goods correctly identify a Werewolf in the speech | $+2$ |
| the Goods wrongly identify a Werewolf in the speech | $-2$ |
| the Goods correctly identify a Good role in the speech | $+1$ |
| the Goods wrongly identify a Good role in the speech | $-1$ |
| Any player who claims that he is a Good role | $+0.5$ |
| # Action-Speech correlated reward | |
| the Seer correctly share his inspection last night | $+2$ |
| the Witch correctly share the usage of antidote or poison | $+1$ |
| any player who claims the voting intention and then vote the same player | $+1$ |
| # Cognition reward | |
| the change $\delta$ of summation of a Werewolf's identity probabilities in the Goods' perspective: | |
| as the Seer | $4\delta$ |
| as the Witch | $2\delta$ |
| as the Hunter or Villagers | $1\delta$ |

### F.5 DETAILS OF OVERALL TRAINING PROCESS

We provide pseudo-code in Algorithm 1; the Thinker and LLMs are trained separately in our framework. This design choice was intentional and serves as one of the strengths of our framework. The separation facilitates training efficiency, as LLMs, which we employ as both Listener and Presenter, inherently generate samples slower than the Thinker module does. Therefore, to optimize our training process, we either employ offline RL or decouple the training between the Thinker and LLMs. The inference workflow is as follows: Listener (LLM) -> language feature $\mathbf{F}$ -> Thinker (RL) -> speech instruction $\mathbf{I}$ -> Presenter (LLM)

During the Thinker's training, the generated speech instructions $\mathbf{I}$ are treated as the new input language features $\mathbf{F}$ for subsequent steps, allowing seamless integration of the RL training into the overall process. Our hybrid training framework incorporates both BC and PPO. During training, each game session is assigned a probability to be either a BC or an RL game. In a BC session, actions $a$ and speaking instructions $\mathbf{I}$ are taken directly from human replays, bypassing Thinker inference. Conversely, in an RL session, the Thinker actively generates actions and speaking instructions. Samples from the game session are tagged as either BC or RL. For the Learner, BC samples utilize the BC loss mentioned in Equation 2, whereas RL samples employ the PPO loss as described in Equation 3.

### F.6 TRAINING HYPER-PARAMETERS

The training hyper-parameters for the Thinker are provided in Table 11.

Regarding the hyperparameters in Equation 4, the Behavioral Cloning coefficient $\alpha$ determines the extent to which the RL policy refers to human strategies as opposed to greedily selecting the RL strategy. We observed that as $\alpha$ decays to 0, Werewolves completely abandon the strategy of claiming to be the Seer due to the high difficulty for Werewolves to convincingly pretend to be the Seer and the relative challenge it poses for RL optimization. A more favorable choice is to masquerade as a villager. Therefore, we maintain a small $\alpha = 0.01$ during the later stages of training, as a constraint for human strategic preferences. Regarding the coefficient $\beta$ for the identity model, we tested values in $1.0, 0.1, 0.01$ and found they had minimal impact on RL, given its nature as an auxiliary learning task. The fine-tuning hyper-parameters for the Listener and Presenter are provided in Table 12.

### F.7 LLM PROMPTING FOR THE LISTENER AND PRESENTER

The information extraction prompt for the Listener module contains the following parts:

- Description of the background of the Werewolf game, as shown in Table 13, which provides the game configuration, game rules, terminology, and descriptions of roles' identities and skills.
- Task requirements, as shown in Table 14. The prompt describes the structured information in JSON format that we expect LLMs to produce, and we describe the appropriate values for each position of the structured command and limit the output within a reasonable range.
- Few-Shot examples, as in Table 16, which provides examples of correctly extracted information from the speeches of different identities and skills, to improve the accuracy of the task as well as to align it with the type of output we expect.
- Current information: Finally, we input the current speech of the player, the game state, e.g., the speaker's *Player id*, role, the current speech types, as in Table 17, to prompt LLMs for deductive reasoning.

The speech generation prompt for the Presenter module comprises the following parts, as shown in Table 18:

- Description of the background of the Werewolf game, which is the same as in the Listener module.
- (Optional) speech instruction. The prompt is a structured output from the Thinker module, and its meaning aligns with that of the Listener module, with a 1-shot example.
- Task requirements, which are similar to those in the Listener module except for the speech generation task.
- Current information, which is similar to that in the Listener module except that we prompt for all historical speeches.

**Algorithm 1:** Pseudo-code for the overall training process.

**Require:**

- Data pairs 1: for fine-tuning of the Listener
  **Input**: [game state $s$, historical speeches $\mathcal{H}$, current player's speech $\mathbf{S}$]
  **Output**: [language feature $\mathbf{F}$]
- Data pairs 2: for finetuning of the Presenter
  **Input**: [game state $s$, historical speeches $\mathcal{H}$, speech instruction $\mathbf{I}$]
  **Output**: [current player's speech $\mathbf{S}$]
- Data pairs 3: for behavioral cloning of the Thinker
  **Input**: [game state $s$, historical collection of all language features $\mathcal{F}$]
  **Output**: [action $a$], or [speech instruction $\mathbf{I}$], decided by the current task type.

---

**Listener and Presenter**:

**if** *use APIs* **then**

    Listener: Use API for generating language features $\mathbf{F}$.

    Presenter: Use API for generating speeches $\mathbf{S}$.

**else**

    Listener: Finetune model with Data pairs 1 and hyperparameters in Table 12.

    Presenter: Finetune model with Data pairs 2 and hyperparameters in Table 12.

---

**Thinker**:

Initialize network parameters for a population of $P$ agents: $\{\theta_1, \theta_2, ..., \theta_P\}$.

Start multiple actors and learners in parallel.

**Actors**: **while** *true* **do**

    Fetch the latest model from the learners. Add the latest checkpoint into a checkpoint list.

    Sample $N - 1$ checkpoints from the list and the latest checkpoint.

    Decide the game episode is BC or RL, run an $N$-player game episode.

    **if** *game episode is BC* **then**

        Get behavioral cloning training samples from Data pairs 3.

    **else**

        Generate RL training samples.

    Accumulate samples in the form $x = (s, \mathcal{F}, a, \mathbf{I}, r, \text{is\_BC})$ and send them to the replay buffer.

**Learners**:

**while** *true* **do**

    **for** $p \in 1, 2, ..., P$ **do**

        Fetch a batch of samples for agent $p$ from the replay buffer.

        Calculate value loss and policy loss according to PPO algorithm in Equation 3.

        Calculate behavioral cloning loss according to Equation 2.

        Calculate loss for auxiliary tasks.

        Update parameters $\theta_p$ using gradients on loss in Equation 4.

Table 11: Hyperparameters for the Thinker training.

| Hyperparameters | Value |
|---|---|
| Population size | 4 |
| Number of actors | 700 (CPUs) |
| Number of learners | 8 (GPUs) |
| Replay buffer size | 100k |
| Mini-batch size | 2048 |
| Max steps | 500k |
| Optimizer | Adam |
| Learning rate | 2e-4 |
| Discount factor ($\gamma$) | 1.0 |
| GAE parameter ($\lambda$) | 0.9 |
| PPO clipping ratio | 0.2 |
| Value function coefficient $c_1$ | 0.5 |
| Entropy coefficient $c_2$ | 0.05 |
| Behavioral Cloning coefficient $\alpha$ | $0.1 \rightarrow 0.01$ |
| Auxiliary task coefficient $\beta$ | 0.1 |

### F.8 GAME LOG EXAMPLES

Table 19 presents a comprehensive analysis of a 9-player werewolf game log, culminating in a victory for the Werewolf.

Table 12: Hyperparameters for fine-tuning the Listener and Presenter.

| Parameter | Listener | Presenter |
|---|---|---|
| # Basic Training Parameters | | |
| Learning rate | 1e-4 | 1e-4 |
| Sequence length | 4096 | 8192 |
| Optimizer | AdamW | AdamW |
| Adam beta1 | 0.9 | 0.9 |
| Adam beta2 | 0.999 | 0.999 |
| Adam epsilon | 1e-8 | 1e-8 |
| Train batch size | 32 | 8 |
| Train epochs | 3 | 3 |
| Max steps | 5000 | 10000 |
| Warmup steps | 500 | 1000 |
| Max grad norm | 1.0 | 1.0 |
| # Model Configuration | | |
| Hidden size | 4096 | |
| KV channels | 128 | |
| Num layers | 28 | |
| Num attention heads | 32 | |
| Layer norm epsilon | 1e-5 | |
| Torch dtype | float16 | |
| # Distributed Training Settings | | |
| Number of GPUs | 8 | |
| Number of nodes | 1 | |
| TP size | 2 | |
| PP size | 1 | |
| # Attention Mechanism Configuration | | |
| Multi query attention | True | |
| Multi query group num | 2 | |

```
                     Werewolf Game Background Prompt

 # Task Scenario:  9-player Werewolf game speech.

 "Good" Faction:
 - 3 Villagers
 - 1 Seer
 - 1 Witch
 - 1 Hunter

 Werewolf Faction:
 - 3 Werewolves

 Common terminologies are explained as follows:
 1.  Werewolf, bandit, wolf, bad faction, knife:  Werewolf.
 2.  Villager, civilian, white card:  Villager.
 3.  Seer, prophet:  Seer.
 4.  Witch, witch card:  Witch.
 5.  Hunter, gun:  Hunter.
 6.  Gold, gold water, verified Good:  A good person verified by the
 Seer.
 7.  Verify Kill:  A Werewolf verified by the Seer.
 8.  Silver, silver water, Werewolves' target, Saved:  A person saved by
 the Witch.
 9.  Iron, steel, certain:  Very certain, e.g., "Player 3 is an iron
 Werewolf" or "Player 3 is definitely the Werewolf," indicates that
 Player 3 is certainly a Werewolf.
 10.  Jump:  A player declares his/her role (not necessarily his/her
 true role).
 11.  Backstab:  A Werewolf sides with the good people, betraying their
 own teammates.
 12.  Defame:  To demean the identity of other players.
 13.  Exalt:  To believe in the identity of other players.
 14.  Vote out, point, nominate, ballot:  Voting, e.g., "Vote for Player
 6 or Player 7," means to vote Player 6 or Player 7 out.
```

Table 13: Werewolf game background prompt.

```
                    Speech Understanding Requirements Prompt

  # Task requirements are as follows:

  Based on your understanding of the game state and speeches, please
  output the extraction results in JSON format in sequence.  The format
  should be:
  {
      "identities": {"<identity>": [player,player,...]} ,
      "actions": {"<action>": [subject player -> object player,
                  subject player -> object player]}
  }

  Example:
  {
      "identities": {"werewolf":[3,5]}, {"<action>": [subject player
      -> object player, subject player -> object player]}, }
      "actions": {"check":[1->6, 2->3]}
  }
  - This indicates Players 3 and 5 are Werewolves, Player 1 checks Player
  6, and Player 2 checks Player 3.
  - Player numbers can only be:  1, 2, 3, 4, 5, 6, 7, 8, 9.
  - When players express their intentions, please correspond to the
  identity of the player, for example, if Player 5 speaks, then consider
  from the perspective of Player 5.
  - The subject number should be inferred from the context, such as
  'I', 'you', 'he', 'she', etc.  If unknown, use 'unknown', for example:
  "check":[unknown->6].

  Possible JSON KEYs are:

  Identities:
  - Roles:  Seer, Witch, Hunter, Villager, Werewolf, "Good" faction,
  Werewolf faction, gold water, silver water, the Werewolves' target,
  etc.
  - Guess:  suspicious, credible, uncertain, tolerant, etc.
  - Speech:  good (up), bad (down), listen well, listen to kill, etc.
  - Faction:  allied, support, werewolf candidate, etc.
  - Online status:  disconnected, offline, not online, voice, etc.

  Actions:
    - Skills:
      - Seer:  check, inspect.
      - Witch:  poison, save.
      - Hunter:  shoot, take away, crash, kill.
      - Werewolf:  self-destruct, explode.
    - And skills that will be used in the future:
      - Vote:  vote out, choose a target, etc.

  - Quotes from other Players' statements do not need to be summarized.
  - Note the distinction between quantifiers and player numbers:  must be,
  that there are three Werewolves.
  - Note negative statements:  not, impossible, implausible, not quite,
  etc.
  - Note the abbreviation of number + information, e.g., "three golds,
  nine slashes, one, six, eight, three wolves" results in:"identities":
  "gold water":[3], "slash":[9], "werewolf":[1,6,8]
```

Table 14: Speech understanding requirements prompt.

```
                        Information Extraction Few-Shot Prompt

   # The following are 11 speeches and corresponding information
   extraction examples:

   Player 3 spoke:  "I checked Player 6, and I suggest Player 8 turn
   around and vote for Player 6.  I will check the identity of Player 4
   in the next round."
   {
       "identities":{"seer":[3],"werewolf":[1,6,8]},
       "actions":{"check":[3->6],"suggest to vote":[8->6],
                  "check in the next round":[3->4]}
   }

   Player 7 spoke:  "Player 2 and I are collaboratively searching for
   a Seer.  Player 2 assists the good faction in combating Werewolves.
   There's a possibility that Player 9 is a werewolf, although I am not
   certain.  The behavior of Player 9 seems suspiciously similar to that
   of Player 2, who possesses the ability to shoot.  Additionally, Player
   4 is identified as a Witch.  Regarding the usage of silver water, I
   suggest targeting Player 6."
   {
       "identities":{"maybe a wolf":[9],"hunter":[2],"silver water":[4]},
       "actions":{"suggest to vote":[7->6]}
   }

   Player 9 spoke:  "Player 8 is the gold water.  Player 2 is not a
   werewolf, neither is Player 3.  However, Player 7 is suspicious, and
   I recommend voting against Player 7.  The roles of Player 4 and Player
   5 are unclear, and Player 1 suspects both of them to be Werewolves.  I
   advise Player 7 to use poison, which could help confirm my role as a
   Seer.  Concerning the hunter, there is a standoff between Player 8 and
   myself.  If there is any uncertainty about Players 1, 2, or 4, the gun
   should be used in this situation against Player 2.  Now, it's time for
   Players 4 and 7 to present their arguments, and there is no need to
   focus on Player 9."
   {
       "identities":{"gold water":[8],"good camp":[2,3],"suspicious":[7],
                     "werewolf":[4,5],"seer":[9] ,"werewolf candidate":
                     [1,2,4], "hunter":[2], "debate players":[4,7]},
       "actions":{"suggest to vote":[9->7],"suggest to poison":[unknown->7]}
   }

   Player 3 spoke:  "Being the first player to speak, my turn was
   conveniently arranged.  However, I am uncertain about Player 2's
   allegiance.  In my view, Player 2 lacks credibility."
   {
       "identities":{"no result": []},
       "actions":{"no result": []}
   }

   Player 7 spoke:  "Player 3 will be poisoned tonight.  I hold the
   Witch card.  I heed the guidance of the two players with gold cards.
   Players 9 and 5 are identified as wolves.  Players 4 and 6 hold cards
   corresponding to their numbers, with Player 4 being more trustworthy
   than Player 5.  Player 3 cannot be revived.  To preserve my own safety,
   I will reveal myself as the Witch.  I have already used the silver
   water card on Player 1.  Player 9 remarked that I should be pleased
   with this misfortune, indicating that the prime werewolf card was
   passed to a fellow teammate."
```

```
{
    "identities":{"witch":[7],"gold water":[2],"werewolf":[9,5],
                  "suspicious":[4]},
    "actions":{"suggest to poison":[7->3],
               "believe to be a silver water":[7->1]}
}

Player 8 spoke:  "Player 5 appears highly suspicious.  He could either
be a werewolf or might be deceiving his teammates.  His failure to set
wolf traps, dishonesty about the wheat sequence, and excessive talking
during the first microphone turn is concerning.  Players 6 and 7 might
be superficial wolves.  Player 7, however, seems to have a sensible
perspective and could be part of the good camp.  I recommend voting
against Player 5."
{
    "identities":{"suspicious":[5],"werewolf":[6,7],"good camp":[7]},
    "actions":{"suggest to vote":[8->5]}
}

Player 2 spoke:  "Regarding the game, my suspicion falls on Players 1,
5, 7, and 3 as potential wolves.  The accusation by Player 3, however,
is incorrect.  I find Player 3's judgment flawed.  It's frustrating.
Similarly, I suspect that Players 1, 5, 7, and 3 are wolves according
to Player 5's perspective.  Let's test this theory.  I propose we
eliminate Player 5 today, and then I, as a Witch, will poison Player
7 tomorrow night.  Observe the game's progression tomorrow, and you
will see that both Player 5 and I, as Witches, agree on Player 2, and
our views align with Player 3's decision.  Therefore, I request that we
focus on Player 5 first."
{
    "identities":{"werewolves' target":[3],"werewolf":[1,5,7],
                  "witch":[2]},
    "actions":{"suggest to vote":[2->5, 2->7]}
}

Player 1 spoke:  "Player 6 is engaging in killing actions.  Players 5
and 7 have been poisoned.  Players 4 and 5 are both targeting Player 1.
Player 3 has been stabbed, and it's possible that Players 2, 4, and 9
each represent a threat, akin to three knives.  Player 5 has revealed
themselves as the Witch and has provided Player 3 with a dose of silver
water."
{
    "identities":{"seer":[1],"poison":[5,7],"depreciate":[4,5],
                  "werewolves' target":[3],"werewolf":[2, 4,9],
                  "witch":[5]},
    "actions":{"check":[1->6],"believe to be a silver water":[5->3]}
}

Player 1 spoke:  "I, Player 1, am part of the good faction.  The focus
of today's game is on Players 3 and 5.  Player 9 might be a werewolf.
I did not use any poison last night."
{
    "identities":{"good camp":[1],"werewolf":[9]},
    "actions":{"suggest to vote":[1->3,1->5]}
}
```

Table 15: Information extraction few-shot prompt.

```
Player 9 spoke:  "I am the Hunter.  Player 7 has self-destructed.
Player 2 might be associated with the silver water.  As for myself,
I reiterate that I am the Hunter.  Player 1 is acting suspiciously,
resembling a white card.  I request the Witch to acknowledge
this.  Player 3 is overly concerned with external cards, which is
uncharacteristic of a Prophet.  Players 3 and 8, please return to the
game, as there's still an opportunity for a round of confrontation."
{
    "identities":{"hunter":[9],"self-destruction":[7],
                  "silver water and seer":[2],
                  "white":[1],"not like a seer":[3] },
    "actions":{"suggest to vote":[9->3,9->8]}
}

Player 4 spoke:  "I believe Player 6 is trustworthy as he revealed
Player 6's key card.  My intention is to verify Player 3.  Player 7,
who holds the gold water, should cast their vote against Player 8.
It's evident that Players 3 and 7 are not the same individual.  On
the field, there are only two players acting as villagers.  I have
identified the three wolves.  There is no necessity to doubt Player
7; instead, Player 4 can be acknowledged as the Seer."
{
    "identities":{"gold water":[7],"seer":[4]},
    "actions":{"consider credible":[4->6],"verified":[4->3],
               "suggest to vote":[4->8]}
}
```

Table 16: Information extraction few-shot prompt.

```
LLM prompting for the Listener

# Task type:  Information Extraction

${{ Werewolf Game Background Prompt }}
${{ Speech Understanding Requirements Prompt }}
${{ Information Extraction Few-Shot Prompt }}
# The task text is as follows:

Player 8 spoke:  "I think Player 9 is a good person, but I am not sure
about the identities of Player 5 and Player 6."

Please directly output the information extraction result in JSON
format:
```

Table 17: LLM prompting for the Listener.

```
                          Speech Generation Prompt

  Now that you play as a Werewolf player, I'm going to provide you with
  some information about the position you're about to speak in, which
  hasn't happened yet and is not historical information, and ask you to
  concatenate this information to generate a paragraph of speech text.

  First, I'll give you some background on the game:

  # Task type:  Game Dialog Generation

  ${{ Werewolf Game Background Prompt }}
  You are playing a 9-player werewolf game.  Suppose you're game Player 1,
  and your identity is Seer.

  I provide you with the format of the in-field message:
  {
      "identities": {"<identity>": [player,player,...]} ,
      "actions": {"<action>": [[subject player, object player],
                  [subject player, object player]]}
  }

  Example:
  {
      "identities": {"werewolf":[3,5]}
      "actions": {"check":[[1,6],[2,3]]}
  }

  - Indicate that Player 3 and Player 5 are Werewolves, Player 1 checks
  Player 6, Player 2 checks Player 3, and the subject and object are
  irreversible.
  - The only possible player IDs are 1,2,3,4,5,6,7,8,9, and unknown should
  be replaced by the speaker's player ID.

  Note that the generated speech result should strictly fulfill the
  following 10 requirements:
  1.  Include all the information in the information extraction result.
  2.  Don't over-imagine and introduce hallucination, and prioritize the
  accuracy of the information.
  3.  The logic between the generated results should be in line with
  the position of the players in Wolfsbane, and there should not be any
  contradictions between the logic before and after.
  4.  Pay attention to the diversity of generated results.
  5.  The generated results should be as anthropomorphic as possible,
  imitating the speaking style of human players.
  6.  Please be firm in your belief that you are the Good faction,
  whether you yourself are in the Good faction or the Werewolf faction.
  7.  Identities or actions can be left out if the result is empty, empty
  is invalid information.
  8.  A player can only be one of the roles of Villager, Seer, Witch,
  Hunter, or Werewolf, for example, it's impossible to be a Witch and a
  Hunter at the same time, if there is more than one conflicting Werewolf
  identity in the information I've provided you with, please randomly
  choose one.
  9.  Please state your identity first.
  10.  Do not make statements such as "unite", "stay alert", "defend the
  village", "together we will go to victory", "find out the werewolf
  as soon as possible", "keep watching", "the information shows up",
  "hopefully we will find it", "think about what we should do next",
  "keep discussing" or similar statements.
```

```
Now that you are Werewolf Player 1, with the identity of Seer, I am
providing you with the key information for your upcoming statement:
{
    "identities": {"villager": [1]},
    "actions": {"no result": []}
}

Please follow the key messages to the letter and keep the text under
100 words:
{
    "identities": {"villager": [1]},
    "actions": {"no result": []}
}

Now it's your turn to speak:
```

Table 18: Speech generation prompt.

```
                        Werewolf Game Log Example

  game role assignment:
 - Player 1:  Werewolf.
 - Player 2:  Witch.
 - Player 3:  Villager.
 - Player 4:  Werewolf.
 - Player 5:  Villager.
 - Player 6:  Hunter.
 - Player 7:  Villager.
 - Player 8:  Werewolf.
 - Player 9:  Werewolf.

  day 1, night:
 - Werewolves:  Player 4, Player 8, and Player 9 choose to kill Player 5.
 - Seer:  Player 1 investigates Player 7, who is in the good camp.
 - Witch:  the werewolf's target is Player 5, and the witch chooses to
 use a potion to save Player 5.

  day 1, announcement:
 - no players were out last night.
 - remaining players:  Player 1(Werewolf), Player 2(Witch), Player
 3(Villager), Player 4(Werewolf), Player 5(Villager), Player 6(Hunter),
 Player 7(Villager), Player 8(Werewolf), Player 9(Werewolf).

  day 1, discussion:
 - (random, Player 9 starts to speak)
 - day 1 Player 9 (Werewolf) said:  "Good man card, ah, the first
 position to speak is not my specialty, you say the system and gave me
 this right, so that I can openly paddling, hey, I especially want to
 exercise, I will say Player 2, just more than a sentence do not say it,
 OK? On the first day to find the next day we in that what in the look
 at the side of the station to update the speech, okay?  Player 9 is a
 white ah, Player 9 Player 8 license plate"
 - day 1 Player 1 (Seer) said:  "The Player 7 is gold water, with a sense
 of hearing, to lead the good guys to victory.  My last words are to
 keep an eye on Player 8 and Player 9 as they are close friends."
 - day 1 Player 2 (Witch) said:  "Player 7 has an average sense of
 hearing and cannot take sides.  Player 1 listens to the situation and
 listens to the speech.  Player 7 only listens to Player 8 or Player 9,
 do the other cards enter the perspective?  Player 5 is a silver water
 cutter, I saved him, don't hold a grudge.  Before Player 6, Player7
 and Player 8 Werewolves should not be less, can go to the highest
 identity."
 - day 1 Player 3 (Villager) said:  "I think Player 4 need be
 intimidated, Player 7 has no sincerity, Player 9's speech is error
 tolerant, Player 2 keeps taking the Witch card, and he's good."
 - day 1 Player 4 (Werewolf) said:  "I'll give gold water on Player 2,
 and Player 9 is labeled as a werewolf.  Player 7 is a predecessor of
 the Player 5 and Player 2 Werewolves.  Player 3 will not be a werewolf.
 Witch card Player 2 doesn't report silver water, which is kind of
 cool.  I'll go ahead and vote for Player 1, with a high probability
 that Player 9 is a werewolf."
 - day 1 Player 5 (Villager) said:  "Player 4 poison Player 1, no need
 for a Seer.  Player 5 has a sense of hearing, Player 7 just doesn't
 fight because he respects his predecessor.  Player 2 is not a one-burst,
 Player 9 must be saved.  Convinced by virtue, vote Player 4 poison
 Player 1 or Player 2 are marked."
```

```
 - day 1 Player 6 (Hunter) said:  "Very well, great, Player 4 feel the
back side does not need to talk about what too much, Player 2 Seer do
not want it, right?  If it's a werewolf, I'm sure I'd want neither."
 - day 1 Player 7 (Villager) said:  "Deacon Player 3 is a white, Player
6 is a top werewolf, Player 4, Player 6, Player 8 three Werewolves,
Player 1, Player 6, Player 9 are numbered cards, Player 4 is a werewolf,
Player 7 suggests betting on Player 6, and the ticket dies poison
Player 1."
 - day 1 Player 8 (Werewolf) said:  "Player 8 questions Player 1's board
and thinks he has intentions.  Player 1's statement mentions checking
Player 7, but it doesn't sound good.  Player 4's statement is off and
suggests washing the oil and playing PK table water tomorrow."

 day 1, voting:
 - voting information:  1 voted for 1, 2 voted for 1, 3 voted for 1,
player 4 abstained, 5 voted for 4, 6 voted for 4, 7 voted for 4, 8
voted for 4, 9 voted for 4
 - voting result:  Player 4
 - remaining players:  Player 1(Werewolf), Player 2(Witch), Player
3(Villager), Player 5(Villager), Player 6(Hunter), Player 7(Villager),
Player 8(Werewolf), Player 9(Werewolf).

 day 1, last words:
 Player 4 (Werewolf) Last Words:  "We should vote Player 5 out and
poison one werewolf, I don't care if I vote them all out.  Player 1
poisoned Player 2, Player 2 witch tags down to find the Hunter.  The
Werewolves may kill Player 2 at night, the gun card should hide.  Witch
card not down tomorrow, you lead the team down, gun card hidden knife
hook."

 day 2, night:
 - Werewolves:  Player 4, Player 8, and Player 9 choose to kill Player 2.
 - Seer:  Player 1 investigates Player 3, who is in the good camp.
 - Witch:  Player 2 poisoned Player 1.

 day 2, announcement:
 - players who died last night:  Player 1, Player 2
 - remaining players:  Player 3(Villager), Player 5(Villager), Player
6(Hunter), Player 7(Villager), Player 8(Werewolf), Player 9(Werewolf).

 day 2, discussion:
 - start from the right of the dead player, Player 3 starts to speak)
 - day 2 Player 3 (Villager) said:  "Player 4 is a true pre-card, Player
5 offers to vote Player 4 poison Player 1, 5, 6, 7 and 8, there is
no opposition, Player7 also has a familiar moderation to him, Player
4'attitude is there, it is not difficult to find a prophet.  Player 6
played Player 4, 6 and 8 in the werewolf pit, Player 6 and Player 7
played awkwardly in the werewolf pit.  I think Player 4 is the true
Seer, and Player 1 is a werewolf card that goes to silver water.  Vote
Player 4 today, and Player 7 says vote Player 4 and Player 8, where is
team Player 4 rolling?"
 - day 2 Player 5 (Villagers) said:  "Right ah, you can hear out the
Player 4 times the Seer, and other people can not hear out ah, on this
chat he wrote to write he went out what is the problem ah?  Eight some
words should not be put on this field to say the end, you can go to say
well, seven or eight shun down well, hey, over."
 - day 2 Player 6 (Hunter) said:  "The Seer points, the next seven
is finished.  Seven doesn't know what the card is and is completely
unintelligible.  Ticket four poison one poison six dry six.  Card seven
must be a werewolf card."
 - day 2 Player 7 (Villager) said:  "Seven is a polearm sign, four is
probably a wolf pre, and six is probably a werewolf.  Ticket seven, but
I would go for ticket six.  There is a white card and a werewolf card
in eight or nine."
```

```
 - day 2 Player 8 (Werewolf) said:  "Player 8 suggests racking silver
 water and hammering Seven for a deep push on Werewolf.  Player 6 and
 Player 9 are two Werewolves' targets, and Player 6 is a werewolf.
 Player 1 and Player 7 are two Werewolves, Player 7 could be the
 werewolf.  Player 9 spoke first, two white cards in Player 7 and Player
 8 in the back seat, and all dumped gold in the front seat.  Player
 6 is the most bandit to speak, and is killed by listening.  It is
 recommended that Player 6 be ordered off first."
 - day 2 Player 9 (Werewolf) said:  "Player 7 and Player 8 are a double
 to play Player 6.  Player 5 has returned to the ticket, and the Player
 7 and Player 8 goes down to a point Player 7.  Player 6 is a white card,
 Player 7 and Player 8 are sold."

 day 2, voting:
 - voting information:  3 votes for 7, 5 votes for 7, 6 votes for 7, 7
 votes for 6, 8 votes for 6, 9 votes for 7
 - voting result:  Player 7
 - remaining players:  Player 3(Villager), Player 5(Villager), Player
 6(Hunter), Player 8(Werewolf), Player 9(Werewolf).

 day 2, last words:
 Player 7 (Villager) Last Words:  "Vote out or poison Player 4, Player
 1 has the same perspective as Player 5, and what was dished out on
 Player 6?  Player 3 may be a werewolf.  Player 2 is a werewolf, leaving
 a three-in-two pattern inside the Player 8 and Player 9, and placing
 Player 3.  Player 9 could be a werewolf."

 day 3, night:
 - Werewolves:  Player 8, and Player 9 choose to kill Player 5.
 - Seer:  the Seer is out.
 - Witch:  the Witch is out.

 day 3, announcement:
 - players who died last night:  Player 5
 - remaining players:  Player 3(Villager), Player 6(Hunter), Player
 8(Werewolf), Player 9(Werewolf).

 day 3, discussion:
 - start from the left of the dead player, Player 3 starts to speak)
 - day 3 Player 3 (Villager) said:  "I do not know whether the double
 wolf has control of the field ah, we are empty, randomly on the three
 ends of the inside point it ah, there is no empty field on the point
 of eight, because I suddenly thought of this nine last speech is too
 brief."
 - day 3 Player 9 (Werewolf) said:
   - policy intent:
 {
     "identities": {"werewolf": [8]},
     "actions": {"suggest to vote": [[9,6]]}
 }
   - [GPT-3.5-LtM]: "On the first day, I quietly observed everyone's
 speeches, and on the second day, the debates heated up.  It seems it's
 time to show my teeth.  First, it must be acknowledged that Player
 5 is a villager; our fellow Werewolves have already eliminated him.
 Judging from the voting results, Player 4 is a suspicious werewolf, and
 we should keep an eye on him.  Additionally, there was something off
 about Player 3's speech; he might be a werewolf too.  Let's pay close
 attention to his movements and not give him a chance to escape.  Now, I
 hope everyone can focus and take down the Werewolves one by one!  "
```

```
   – [GPT-3.5-T]: "I am Player 9, and I sincerely tell everyone that I
am a villager.  Based on the statements and votes of the past two days,
I believe Player 6 is a werewolf, while Player 8 and I are villagers.
We can cast our votes for Player 6, because his actions and words
have made me feel that he is a werewolf.  Let's stay vigilant and work
together to find the real werewolf to ensure the peace of our village.
Thank you all for listening."

   – [GPT-4-LtM]: "I am Player 9.  Last night, we lost Player 5, which
is really unfortunate.  The situation is currently complex, but
I believe that Player 3 is likely a werewolf.  His speeches seem
insincere, and he seems to hold back when analyzing the game situation.
Additionally, based on the analysis of yesterday's last words, I think
Player 8 is also very suspicious.  He might be promoting actions that
are advantageous to the Werewolves.  Therefore, I suggest that today
we consider voting for either Player 3 or Player 8.  I have a strong
intuition that one of them is a werewolf."

   – [GPT-4-T]: "As Player 9, I'm just an ordinary villager.  The
situation is becoming clearer by the moment.  From my perspective,
Player 8 is definitely a werewolf--it's a done deal.  This also
explains why I strongly suggest that we vote for Player 6 today.
Stating one's position clearly is important.  In my view, Player 6's
speeches and actions have raised many suspicions, which is the main
reason I find him untrustworthy.  Therefore, I will cast my vote for
Player 6 and hope for your support to ensure that we can identify the
real Werewolves.  "

   – [WereLLM-T]: "I am Player 9.  I think Player 8 might be a werewolf
because his statements aren't very good.  He tries to distract by
targeting Player 6, and although Player 7's statements are okay, they
don't provide useful information.  I also suspect that Player 6 might
be a werewolf, so I will vote for him.  "

 – day 3 Player 8 (Werewolf) said:  "Can't kill Player 8 anymore, Player
8 is a werewolf come gun hang yourself."
 – day 3 Player 6 (Hunter) said:  "Player 6 is a Werewolf, step on."

 day 3, voting:
 – voting information:  3 abstained, 6 voted for 6, 8 abstained, 9 voted
for 6.
 – voting result:  Player 6

 game result:
 the Werewolves win the game.  (final surviving player:  Player 3
(Villager), Player 8 (Werewolf), Player 9 (Werewolf))
```

Table 19: Werewolf game log example.

