# OpenReview forum: "Enhance Reasoning for Large Language Models with Reinforcement Learning in the Game Werewolf"
_ICLR.cc/2025/Conference — Submitted to ICLR 2025_

### Official Review · Reviewer_ShgQ · 2024-11-01

**Soundness:** 3
**Presentation:** 3
**Contribution:** 2
**Rating:** 5
**Confidence:** 4

**Summary:**

The paper introduces an inventive framework to enhance LLM's reasoning abilities, consisting of domain-agnostic System-1 and Thinker module focuses on System-2 tasks that require complex logical analysis and domain-specific knowledge. The results show that this integrated approach significantly improves the LLMs' deductive reasoning, strategic planning, and alignment with human-like decision-making in the Werewolf game context.

**Strengths:**

1. The paper introduces an innovative framework that enhances the reasoning capabilities of large language models (LLMs) by integrating a specialized "Thinker" module.
2. The creation and utilization of the largest dataset for social deduction games, consisting of 18,800 real human game sessions, not only provides a solid basis for the study’s conclusions but also offers a valuable resource for future research.
3. The paper is well-structured, with clear sections delineating the problem area, the novel contributions, the methodology, and the findings.

**Weaknesses:**

1. The framework is highly specialized for the Werewolf game, which may limit its generalizability to other domains or types of reasoning tasks.

**Questions:**

1. In part"Online evaluation win rates with 1 human and 8AIs.", you can add more human to do the online experiment to provide more sounding results, and add GPT-4(w/o any external module) into the experiments so that we can get the effect of improvement of the three modules.
2. I don't know in figure 3, that why part "Vote Werewolves" illustrates that in day2 and day3, GPT-4-LtM performs worse than GPT-4, which is converse to other parts. Maybe you can analyze the error reason via the game history.

---

### Official Review · Reviewer_Vx1D · 2024-11-03

**Soundness:** 2
**Presentation:** 2
**Contribution:** 2
**Rating:** 3
**Confidence:** 4

**Summary:**

The paper addresses the challenge of enhancing reasoning in LLMs when they are used in domain-specific tasks not well-covered in their pre-training data. The authors propose an innovative framework that integrates a general-purpose LLM with an external "Thinker" module, which directly accesses domain knowledge and is trained using RL and supervised learning. This dual-process framework assigns intuitive tasks to the LLM and complex, analytical tasks to the Thinker.
The framework is tested on a 9-player version of the social deduction game Werewolf, where LLMs handle NLP tasks while the Thinker solves complex reasoning tasks, such as strategic planning and role deduction.

Key contributions:

* A dual-process framework where an external Thinker module augments LLM reasoning capabilities.
* Dataset for social deduction games to date, intended to support future research in this area.
* Demonstrated improvements in LLM performance for deductive reasoning and social deduction tasks.

**Strengths:**

1. Datasets: FanLang-9, an interesting dataset for reasoning field.

**Weaknesses:**

1. The framework is only designed for Werewolf or FanLang-9. I am not convinced by its applicability to broader reasoning tasks. The domain-specific customization of the Thinker for Werewolf limits its generalizability, especially in tasks requiring different types of strategic thinking or non-social contexts.
2.  No other benchmark/dataset is used for evaluation. The paper focus on the framework not the dataset itself. So current experiments are not enough to prove that the ``Experiments show that GPT-3.5 and GPT-4, augmented with the Thinker, significantly improve in deductive reasoning, textual speech generation, and online gameplay evaluated by human players.''
3. Baselines: These baselines are not quite fair. There are many agents LLM which has quite similar structures.

**Questions:**

1. RL: In the paper, ``...The generation of a speech instruction can be viewed as a multi-label classification problem and decomposed into multiple single-class classifications for each attribute convert it into N × M discrete actions and apply the identical training algorithm...". How to apply this to other field or tasks? Is your framework designed only for one game? For open-ended questions/tasks, what are the solutions here?

---

### Official Review · Reviewer_zRPo · 2024-11-04

**Soundness:** 2
**Presentation:** 3
**Contribution:** 2
**Rating:** 5
**Confidence:** 4

**Summary:**

This paper introduces a framework that enhances reasoning in large language models (LLMs) by integrating them with an external Thinker module. The Thinker module is optimized through reinforcement learning on domain specific dataset, and is designed to handle complex System-2 reasoning tasks requiring domain-specific logic. The thinker module works alongside LLMs for intuitive System-1 tasks requiring general, domain-agnostic language generation. This approach is demonstrated in the context of the Werewolf social deduction game. The experiments show that the Thinker enables more effective strategic reasoning and decision-making. They've also collected a dataset of 18,800 human games in Werewolf for domain specific training.

**Strengths:**

- Instead of working purely on LLMs (prompt engineering, finetuning, etc.), the authors demonstrate that it's possible to train an external module with reinforcement learning to enhance the reasoning of LLMs. Their thinker module is light-weighted and easy to train;
- The authors collected a dataset of 18,800 human Werewolf games, including both game states and audio data, which equates to around 7000 hours of gameplay and 6000 hours of audio. This dataset may encourage further research in areas like social reasoning and conversational AI;
- Experiments demonstrate that the proposed framework outperforms baseline methods, including models using advanced prompt engineering. The Thinker-augmented system achieves impressive gains in deductive reasoning accuracy, speech generation quality, and overall gameplay performance, indicating the framework’s effectiveness;

**Weaknesses:**

- Although the authors claim their method diverges from Cicero in several aspects, I'm still concerned that this work may lack novelty. The authors claim that "actions in Cicero require both NLU and NLG involves a high-level and task-related reasoning beyond domain-agnostic NLP" (I also believe there's some grammar mistake in here). But their framework also involves both NLU and NLG -- they interact with an external Thinker module instead of within an LLM. The authors also claim that Cicero necessitates fine-tuning of LLMs, but they didn't demonstrate if further finetuning their LLMs can further enhance performance or not.
- The statement of the human evaluators involved in the experiments lacks clarity. The statement, 'we recruited 10 human evaluators, all well-versed in the Werewolf game,' does not specify what qualifies them as 'well-versed.' This vague term may lead to potential bias in the experimental results. A ground truth test on the human evaluators would strengthen the validity of the evaluation.
- As the authors mention in their limitation that "Our evaluations primarily involved games featuring either AI vs AI or one human player competing against multiple AIs. Evaluating an AI in a majority-human player setting presents challenges due to the highly interactive nature of the game and the variability in human players’ speech strategies and behaviors", the lack of results from direct comparison between the AI and a larger group of human players makes the results less promising. This limitation restricts our understanding of the AI's performance and adaptability in realistic, diverse human interactions.

**Questions:**

1. Can the authors clarify how their framework’s reliance on both NLU and NLG fundamentally diverges from Cicero, aside from the integration of the Thinker module?
2. Would the authors consider testing whether further fine-tuning of their LLMs could improve performance? If this has been explored, what were the findings?
3. Could you provide a more detailed description of the human evaluators in your experiments? For example, do you have a test for these evaluators to see how accurate they are on some ground truth dataset?

---

### Meta-Review · Area_Chair_Gate · 2024-12-18

**Metareview:**

This paper proposes a framework integrating Large Language Models with an external "Thinker" module to enhance reasoning in domain-specific tasks, demonstrated through the social deduction game Werewolf. The framework employs a dual-system design where LLMs handle intuitive tasks and the Thinker focuses on logical analysis. The work introduces a large dataset of 18,800 human games and reports performance improvements in reasoning and gameplay when using the Thinker module.

The reviewers have raised that the work seems overly specialized to the Werewolf game with unclear generalizability to other tasks, and lacks novelty compared to Cicero, as differences are insufficiently justified. Baselines and experimental settings may not be fair or sufficient to validate the claimed improvements. Insufficient clarity and rigor in describing the qualifications of human evaluators and the Thinker module's broader applicability. Overall, this paper requires more work.

**Additional Comments On Reviewer Discussion:**

No discussion from the authors in response to the reviewers.

---

### Decision · Program_Chairs · 2025-01-22

Reject